# TUG1-mediated R-loop resolution at microsatellite loci as a prerequisite for cancer cell proliferation

Miho M. Suzuki [1,12], Kenta Iijima [1,2,12], Koichi Ogami [3], Keiko Shinjo [1], Yoshiteru Murofushi [1], Jingqi Xie [1], Xuebing Wang[1], Yotaro Kitano [4], Akira Mamiya [1], Yuji Kibe[1,4], Tatsunori Nishimura[1], Fumiharu Ohka [4], Ryuta Saito [4], Shinya Sato [5], Junya Kobayashi [6], Ryoji Yao [7], Kanjiro Miyata[8], Kazunori Kataoka [9,10], Hiroshi I. Suzuki [3,11] & Yutaka Kondo [1,11] ✉

Oncogene-induced DNA replication stress (RS) and consequent pathogenic R-loop formation are known to impede S phase progression. Nonetheless, cancer cells continuously proliferate under such high-stressed conditions through incompletely understood mechanisms. Here, we report taurine upregulated gene 1 (TUG1) long noncoding RNA (lncRNA), which is highly expressed in many types of cancers, as an important regulator of intrinsic R-loop in cancer cells. Under RS conditions, TUG1 is rapidly upregulated via activation of the ATR-CHK1 signaling pathway, interacts with RPA and DHX9, and engages in resolving R-loops at certain loci, particularly at the CA repeat microsatellite loci. Depletion of TUG1 leads to overabundant R-loops and enhanced RS, leading to substantial inhibition of tumor growth. Our data reveal a role of TUG1 as molecule important for resolving R-loop accumulation in cancer cells and suggest targeting TUG1 as a potent therapeutic approach for cancer treatment.

DNA replication ensures the precise duplication of genetic information through the cell cycle. Slowing or stalling of DNA replication fork progression, collectively described as "replication stress (RS)", leads to single-stranded DNA (ssDNA) and DNA strand breaks[1]. Although RS is not a common feature of normal cells, failure in the cellular response to RS is a major cause of genome instability and is linked to tumorigenesis; thus, RS is a characteristic feature of cancer[1]. In cancer cells, activated oncogenes accelerate the cell cycle and drive uncontrolled S phase entry that correlates with RS and overabundant R-loop formation[2]. The latter are three-stranded nucleic acid structures

[1]Division of Cancer Biology, Nagoya University Graduate School of Medicine, 65 Tsurumai-cho, Showa-ku, Nagoya, Aichi 466-8550, Japan. [2]Laboratory Animal Facilities and Services, Preeminent Medical Photonics Education and Research Center, Hamamatsu University School of Medicine, 1-20-1 Handayama, Higashi-ku, Hamamatsu, Shizuoka 431-3192, Japan. [3]Division of Molecular Oncology, Nagoya University Graduate School of Medicine, 65 Tsurumai-cho, Showa-ku, Nagoya, Aichi 466-8550, Japan. [4]Department of Neurosurgery, Nagoya University Graduate School of Medicine, 65 Tsurumai-cho, Showa-ku, Nagoya, Aichi 466-8550, Japan. [5]Molecular Pathology and Genetics Division, Kanagawa Cancer Center Research Institute, 2-3-2 Nakao, Asahi-ku, Yokohama, Kanagawa 241-8515, Japan. [6]School of Health Sciences at Narita, International University of Health and Welfare, 4-3 Kozunomori, Narita, Chiba 286-8686, Japan. [7]Department of Cell Biology, Japanese Foundation for Cancer Research, 3-8-31 Ariake, Koto-ku, Tokyo 135-8550, Japan. [8]Department of Materials Engineering, Graduate School of Engineering, The University of Tokyo, 7-3-1 Hongo, Bunkyo-ku, Tokyo 113-8656, Japan. [9]Innovation Center of NanoMedicine, Kawasaki Institute of Industrial Promotion, 3-25-14 Tono-machi, Kawasaki-ku, Kanagawa 210-0821, Japan. [10]Institute for Future Initiatives, The University of Tokyo, 7-3-1 Hongo, Bunkyo-ku, Tokyo 113-0033, Japan. [11]Institute for Glyco-core Research (iGCORE), Tokai National Higher Education and Research System, Furo-cho, Chikusa-ku, Nagoya, Aichi 464-8601, Japan. [12]These authors contributed equally: Miho M. Suzuki, Kenta Iijima. ✉e-mail: ykondo@med.nagoya-u.ac.jp

composed of an RNA/DNA hybrid and non-template ssDNA. Such unscheduled R-loops can block replication fork progression, resulting in amplified RS, which is itself a cause of R-loop formation[3]. Thus, RS and R-loops appear to establish a reinforcing loop and promote genome instability in cancer cells[4].

Because scheduled R-loops are formed at certain specific regions where they mediate specific physiological functions such as immunoglobulin class switching recombination and specific regulatory steps in transcription initiation and termination[5], they need to be meticulously regulated. Cells usually employ multiple factors and mechanisms to stringently prevent the deleterious effects of unscheduled R-loops in normal situations[3]. For example, topoisomerases, such as Topoisomerase I, are thought to prevent accumulation of R-loops by relaxing DNA supercoiling[6]. The Ribonuclease H (RNase H) enzyme specifically cleaves the RNA moiety of the RNA/DNA hybrids in a sequence-independent manner, resulting in the removal of RNA from the DNA[3]. A number of helicases, such as Petite Integration Frequency 1 (PIF1), Sen1/Senataxin (SETX), and Fanconi anemia complementation group M (FANCM), have been shown to unwind RNA/DNA in vitro[5]. Recently, DExH-Box Helicase 9 (DHX9) has been demonstrated to interact with RNA/DNA hybrids and promote R-loop suppression and transcriptional termination in vivo. However, the underlying mechanisms responsible for DHX9 recognition and suppression of unscheduled R-loops remain unclear[7].

ssDNA formed in the R-loop structure is bound by replication protein A (RPA) heterotrimers consisting of the three subunits RPA70, RPA32, and RPA14 known to protect ssDNA from single- and double-strand breaks (DSBs)[8]. A recent study reported that ssDNA at or near R-loop structures allows RPA to sense the increase of genomic stress and help to remove R-loops in front of replication forks via RPA-RNase H1 interactions[8]. However, it is still under investigation as to which mechanisms effectively remove R-loops in global loci in order to maintain the balance between proliferation and abundance of R-loops and RS in cancer cells.

Given that RS and R-loop accumulation are potential sources of hypermutation, chromosomal rearrangements, or chromosome loss that are important in cancer progression and maintenance, it is unsurprising that cancer cells frequently exhibit abundant RS and R-loops as a result of the continuous proliferative signaling and/or the loss of repair system of stressed replication forks[9,10]. However, paradoxically, continuously growing cancer cells must circumvent severe replication failure by efficiently resolving excessive R-loops and RS in order to avoid falling into replication catastrophe[11–13].

Evidence accumulated over the past decade shows that long noncoding RNAs (lncRNAs) regulate gene transcription, chromatin modifications, spatial organization of the genome, genomic stability and DNA damage repair in the nucleus, and the stability of microRNAs (miRNAs) or translational efficiency of mRNAs in the cytoplasm[14–17]. Considering these multiple pivotal roles of lncRNAs in nuclei, and the knowledge that RNA can be a fast-acting molecule for regulating cell functions in response to extracellular signals, lncRNAs may also regulate intrinsic RS via assisting in the removal of R-loops in cancer cells. In the current study, we aimed to identify the role of taurine upregulated gene 1 (TUG1) and investigated its ability to cope with R-loops in cancer cells. Furthermore, we provide evidence that targeting TUG1 enhances DNA damage caused by chemotherapy-induced R-loops[18] which may be amenable to clinical exploitation.

## Results

### Induction of lncRNA TUG1 expression resulting from RS and R-loop accumulation
To identify the lncRNAs upregulated in response to RS and R-loop accumulation, we synchronized HeLa/Fucci2 cells in the S phase by mitotic shake off[19] and induced RS by treatment with hydroxyurea (HU) or camptothecin (CPT). Synchronization and enrichment of cells

in S phase was validated by EdU staining, which specifically detects such cells (Supplementary Fig. 1a). HU stalls replication forks by depleting the dNTP pool[20]. CPT traps Topoisomerase I and increases DNA negative supercoiling, which promotes R-loop accumulation that leads to DSB[21,22]. After two hours of treatment, we extracted RNAs newly transcribed on chromatin (Supplementary Fig. 1b). Of 16,193 annotated lncRNAs in GENCODE v38 (GRCh38.p13), 14 were upregulated more than 2.0-fold either in HU- or CPT-treated cells (Fig. 1a, Supplementary Data 1). TUG1 was common to both treatments, and exhibited the highest degree of upregulation.

Reverse transcription quantitative PCR (RT-qPCR) validated the upregulation of TUG1 in the S phase (Fig. 1b). Single-molecule fluorescence in situ hybridization (smFISH) and qRT-PCR revealed an increase in the number of TUG1 molecules particularly in the nucleus within two hours of treatment with either HU or CPT (Fig. 1c, d, Supplementary Fig. 1c, d). The nuclear increase in TUG1 expression was followed by a cytoplasmic increase after four hours (Fig. 1d). Because the inhibition of transcription by 5,6-dichloro-1-β-D-ribofuranosylbenzimidazole (DRB) impaired the induction of TUG1 by HU or CPT, the increase in nuclear TUG1 was due to new transcription (Fig. 1e). TUG1 was also upregulated predominantly in the S phase but not in the G1 phase by treatment with bleomycin, a cell-cycle independent DNA-damaging agent, suggesting that TUG1 was induced specifically in response to RS (Fig. 1f).

The induction of TUG1 by HU was impaired by the ATR inhibitor VE-821 and the CHK1 inhibitor SCH900776, indicating that the ATR pathway is a major signal transduction pathway for induction of TUG1 (Fig. 1g, h). E2F1 and E2F6 function downstream of ATR-CHK1 as an activator and repressor, respectively, in response to RS[23]. Indeed, we found that recruitment of E2F1 and exclusion of E2F6 from the TUG1 promoter after treatment was closely associated with its expression (Supplementary Fig. 1e, f). The effect of E2Fs on TUG1 expression was further validated by siRNA treatment (Supplementary Fig. 1g). While binding of MYC was observed in the TUG1 promoter, the enrichment level of MYC was unchanged before and after treatment with HU and CPT (Supplementary Fig. 1e, h)[24].

TUG1 is highly expressed in various different cancers relative to their normal tissue counterparts (Supplementary Fig. 1i). Induction of TUG1 by HU was observed in multiple cancer cell lines, including those with wild-type TP53 (U2OS), mutant TP53 (LN229, T98G, U251MG), or TP53 inactivated by HPV-encoded E6 protein (HeLa). This implies that TUG1 induction by RS is not necessarily TP53-dependent[23] (Supplementary Fig. 1j).

### Interaction of TUG1 with pRPA32 and DHX9 at R-loop regions is enhanced by RS and R-loop accumulation
Upon DNA damage, RS and R-loop formation, ssDNAs are bound by the RPA-trimer complex, in which the subunit RPA32 is phosphorylated (e.g. ATR-dependent phosphorylation of RPA32 at Ser33)[8,25]. Super-resolution single-molecule fluorescence microscopy revealed that treatment with either HU or CPT increased the colocalization of phosphorylated RPA32 (pRPA32) and TUG1 in the S phase nucleus, while untreated controls showed various levels of colocalization in each type of cancer cell line tested (Fig. 2a, b, Supplementary Table 1). In contrast, γ-H2AX, a marker of DSB, minimally colocalized with TUG1 two hours after CPT treatment, suggesting that TUG1 preferentially localized at the early stage of RS or R-loop formation where ssDNA was bound to arrays of RPA rather than DSB loci[26] (Fig. 2c, Supplementary Table 1).

The interaction between TUG1 and RPA complexes was further supported by RNA immunoprecipitation (RIP) assays using HeLa and U2OS cells, in which either RPA32 or RPA70 was stably expressed (Fig. 2d, Supplementary Fig. 2a). TUG1 but not control mRNA (hypoxanthine phosphoribosyltransferase, HPRT) strongly bound RPA32 in CPT-treated cells. TUG1 also bound RPA70 in CPT-treated cells, but less well compared with RPA32.

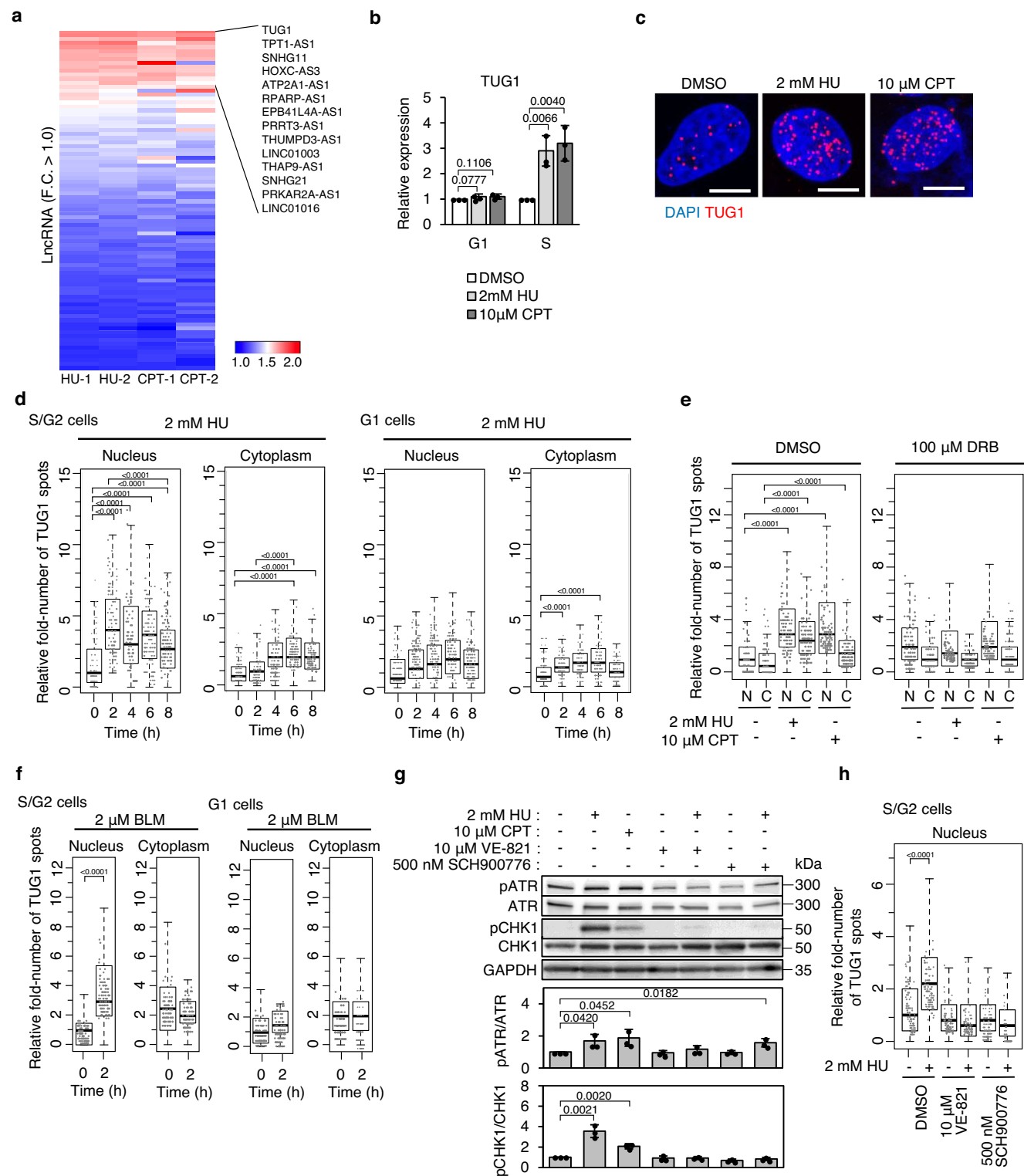

In order to further examine the relationships between TUG1, RPA, and R-loops, we expressed catalytically inactive RNase H1 (RNH1$^{D210N}$-GFP), which can recognize and stably bind to R-loops[8,27–33]. Interestingly, around half of the TUG1 foci clearly colocalized with RNH1$^{D210N}$-GFP foci in CPT-treated cells (Fig. 2e, Supplementary Table 1). Furthermore, we observed striking colocalization of TUG1 and pRPA32 at the R-loop regions (Fig. 2e). Next, we asked whether the colocalization of TUG1, RNH1$^{D210N}$-GFP, and pRPA32 occurred at replication forks. Proliferating cell nuclear antigen (PCNA), which is a co-factor of DNA polymerase-delta[34], colocalized with about half of the TUG1 and RNH1$^{D210N}$-GFP foci (Supplementary Fig. 2b, Supplementary Table 1). Newly synthesized DNA strands labeled with 5-ethynyl-2′-deoxyuridine (EdU) also colocalized with about half of the TUG1 and RNH1$^{D210N}$-GFP foci (Supplementary Fig. 2c, Supplementary Table 1). In view of the fact that TUG1 and RNH1$^{D210N}$-GFP foci colocalize with pRPA32 (Fig. 2e), these data suggest that a certain part of TUG1 is located at the DNA replication site, where transcription-replication conflicts occur.

**Fig. 1 | RS rapidly induces TUG1 expression. a** Heat map representing fold changes in lncRNA expression after HU or CPT treatment. The experiments were conducted in duplicate. 14 lncRNAs with upregulation on average > 2.0-fold of HU-1, HU-2, CPT-1, and CPT-2 are indicated. F. C., fold change. **b** RT-qPCR analyzes of TUG1 expression in cell-cycle-synchronized HeLa/Fucci2 cells treated with DMSO, HU, or CPT for 2 h. Mean ± SD, $n = 3$. Two-sided $t$-test. **c** Representative images of TUG1 smFISH in HeLa/Fucci2 cells treated with DMSO, HU, or CPT for 2 h. Scale bar = 10 μm. **d** Quantification of smFISH experiments. The y-axis indicates the relative fold-number of TUG1 spots in the nucleus and cytoplasm after 0, 2, 4, 6, and 8 h of HU treatment compared with the median number of TUG1 spots in the S/G2 nucleus at 0 h. **e** Inhibition of transcription by DRB suppresses TUG1 induction. HeLa/Fucci2 cells were incubated with DRB for 30 min before treatment with HU or CPT for 2 h. N, nucleus, C, cytoplasm. **f** Relative fold-number of TUG1 spots after

bleomycin (BLM) treatment compared with the median number of TUG1 spots in the S/G2 nucleus of DMSO-treated cells at 0 h. For **d**, **e**, and **f**, more than 60 HeLa/Fucci2 cells were analyzed per sample. Two-tailed Wilcoxon rank sum test. **g** Top, Western blotting of HeLa/Fucci2 cells treated with an inhibitor of ATR (VE-821) or CHK1 (SCH900776) for 30 min before treatment of HU or CPT for 2 h. Bottom, bar graph generated by quantifying the Western blot. Mean ± SD, $n = 3$. Two-sided $t$-test. **h** Relative fold-number of TUG1 spots in cells treated with VE-821 or SCH900776 for 30 min before HU treatment for 2 h. More than 30 cells were analyzed per sample using one-way ANOVA and Tukey's multiple comparison tests. For **d**, **e**, **f**, and **h**, in the box plot, center lines show medians; box limits indicate the 25th and 75th percentiles; whiskers extend 1.5 times the interquartile range from the 25th and 75th percentiles. The experiments were conducted in triplicate with similar results. Source data are provided as a Source Data file.

Dumbovic et al. recently showed that both fully spliced and intron-retained transcripts are present in the nucleus, while only spliced transcripts exist in the cytoplasm[35]. Consistent with their report, we found similar numbers of spliced and intron-retained transcripts in the nucleus by smFISH using a probe set targeting TUG1 exon 2, which can detect both spliced and intron-retained transcripts, and a probe set targeting both intron 1 and intron 2, which detects only intron-retained transcripts. After CPT treatment, RNH1$^{D210N}$-GFP colocalized with both fully spliced and intron-retained transcripts (Supplementary Fig. 2d, Supplementary Table 1).

Next, affinity pull-down assays were performed to identify TUG1-interacting proteins other than RPA complexes under RS, in order to clarify implications of TUG1 for R-loop metabolism. Using biotinylated synthetic full-length TUG1, we identified DHX9, an ATP-dependent RNA helicase A, at the 140 kDa bands by mass spectrometric analysis (Fig. 3a and Supplementary Table 2, Methods). Interactions between synthetic TUG1 and DHX9 were also validated by Western blotting (Fig. 3b). Consistently, super-resolution single-molecule fluorescence microscopy showed colocalization of TUG1 and a part of DHX9 in the S phase nucleus of HeLa cells treated with CPT (Fig. 3c).

Ultraviolet (UV)-crosslinking immunoprecipitation and qPCR (CLIP-qPCR) in vivo assays were performed to identify the binding regions of DHX9 on TUG1. DHX9 bound directly to two separate regions (3-2 and 3-3) within exon 3 of TUG1, probably due to the secondary structure of TUG1 in the cells (Fig. 3d). In contrast, pRPA32 directly interacted with TUG1 at the 5′ of exon 3 (3-1). Consistent with this, the biotinylated RNA pull-down analysis in vitro using a series of partially deleted TUG1 fragments revealed that DHX9 did not interact with the Δ4 fragment (1–6, 119 bp) of TUG1, while it did with Δ3 (1–4, 739 and 6266-7542 bp). These mutants lack region 3-3 and 3-2, respectively, as shown in the CLIP assay. This indicates that the 3′ portion of exon 3 (region 3-3) of TUG1 is dominantly required for interactions with DHX9 (Fig. 3e, Supplementary Fig. 3a). Notably, RNA pull-down analysis of the Δ2 fragment revealed that a 1339−3834 bp region of TUG1 is required to bind RPA32, which contains the RPA32 binding site (3-1) detected by CLIP assay (Fig. 3d,e). The interaction between TUG1 and DHX9, and TUG1 and pRPA32 were also detected by CLIP-qPCR in the four cell lines examined (Supplementary Fig. 3b).

In order to further examine the interplay between TUG1, DHX9, and RPA32 in the native cellular context, we additionally applied the CRISPR-assisted RNA-protein interaction detection method (CARPID), a proximity-labeling-based methodology[36] (Supplementary Fig. 3c). CARPID documented the localization of both DHX9 and RPA32 within a range of 1–10 nm from TUG1 in the nucleus of CPT-treated cells (Supplementary Fig. 3d).

### Depletion of TUG1 increases R-loops and RS in S phase

Clear colocalization of TUG1, pRPA32, and DHX9 shown by super-resolution microscopy may indicate the mechanistic role of TUG1 in R-loop resolution corresponding to the level of RS. We then examined the effect of TUG1 depletion on R-loop metabolism. We used

chemically-modified chimeric DNA antisense oligonucleotides (ASO) for the effective knockdown (KD) of nuclear-retained lncRNA[37]. TUG1 KD by two independent ASOs (referred to as TUG1#1 and TUG1#2) almost completely prevented TUG1 expression within 24 h (<1.0% of control in HeLa/Fucci2, LN229, U251MG, and U2OS and <5.0% of control in T98G) (Fig. 4a, Supplementary Fig. 4a). Substantial (>90%) depletion of TUG1 expression by ASO was achieved even within 2 to 4 h after transfection in the four cell lines examined. The short period of ASO treatment is useful to understand the direct effects of TUG1 KD. Note that the DHX9 protein level was unchanged by TUG1 KD (Supplementary Fig. 4b).

Accumulation of R-loops was examined by slot-blot analysis using the S9.6 antibody, which recognizes RNA/DNA hybrids[38] (Fig. 4b). TUG1 depletion significantly increased R-loop formation in HeLa/Fucci2 cells (4 h of treatment) (Fig. 4b). The S9.6 signal was completely abrogated by RNase H treatment of DNA in vitro, supporting the specificity of S9.6 antibody against RNA/DNA hybrids in this experiment. DRB treatment of the cells also significantly decreased the S9.6 signal, suggesting that they originated from nascent RNA transcription (Fig. 4b).

To evaluate RS, double-pulse labeling with 5-iodo-2′-deoxyuridine (IdU) and 5-chloro-2′-deoxyuridine (CldU) was conducted to measure ongoing replication fork speed at single-molecule resolution in TUG1 KD cells (Fig. 4c, Supplementary Fig. 4c). Four hours after TUG1 KD, the distribution of CldU-labeled tracks shifted towards shorter values. The median fork speed was significantly reduced from 0.91 kb/min in controls to 0.78 kb/min in TUG1-depleted HeLa/Fucci2 cells. This reduction of fork speed was also observed in T98G, U251MG, and U2OS cells (Fig. 4c, Supplementary Table 3). Immunohistochemistry and flow cytometry (FCM) also revealed a significant reduction of EdU incorporation into S phase cells (Supplementary Fig. 4d,e). After 24 h of TUG1 KD, EdU incorporation was more substantially decreased (Supplementary Fig. 4f, g).

In HeLa/Fucci2, the slowed replication fork progression led to a prolonged S phase as reflected by an increased S/G2 cell population[19] from 42 to 65% of all TUG1 KD cells after 24 h (Fig. 4d). S/G2 phase arrest was also observed in other types of cancer cell lines examined (Supplementary Fig. 4h). Taken together, these data indicate that depletion of TUG1 caused both the accumulation of R-loops and RS during S phase in all of the cancer cell lines tested.

### TUG1 resolves R-loops via DHX9 activity

To determine the mechanism of action of TUG1 on R-loop resolution, we used the U2OS 2-6-3/TA manipulated cell line (Fig. 4e, Supplementary Fig. 5a, Methods). In the presence of Doxycycline (Dox), the reverse tetracycline transactivator (rtTA) drives high expression of the cyan fluorescent protein (*CFP*) gene. The inducible *CFP* gene cassette is flanked by a LacO array that can tether TUG1-PP7 via interaction with PP7-binding protein (PCP) fused with LacR. Approximately 200 copies of this moiety are tandemly integrated in a single locus of U2OS 2-6-3/TA cells[39] (Fig. 4e). The induction of *CFP* expression induced R-loop

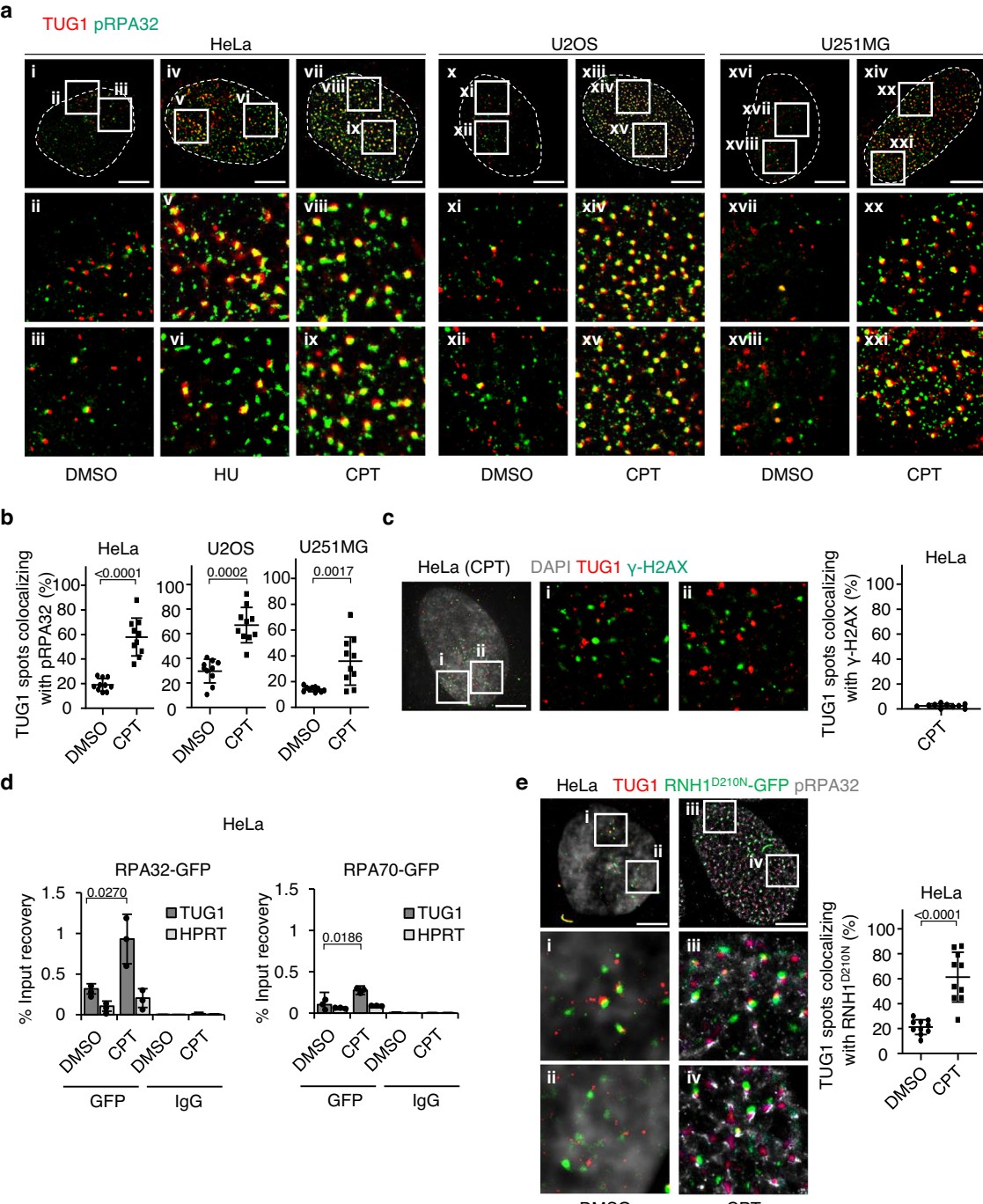

**Fig. 2 | TUG1 interacts with pRPA32 at R-loop. a** Super-resolution images of the nucleus (dotted line) co-stained with pRPA32 (green) and TUG1 smFISH (red). (i-ix) HeLa cell treated with DMSO (untreated control), or 2 mM HU for 2 h, or 10 μM CPT for 2 h. (x-xv) U2OS cell treated with DMSO or 10 μM CPT for 2 h. (xvi-xxi) U251MG cell treated with DMSO or 10 μM CPT for 2 h. Magnified regions where TUG1 and pRPA32 colocalize in the top panels were shown in the bottom two panels. Three independent experiments were carried out with similar results (Supplementary Table 1) and representative images are shown. Scale bar = 5 μm. **b** The percentage of TUG1 spots colocalizing with pRPA32 relative to total number of TUG1 spots in each cell. **c** Left, representative super-resolution images of TUG1 (red) and γ-H2AX (green) in HeLa cell treated with 10 μM CPT for 2 h. Magnified regions in the left panel were shown in middle (i) and right (ii) panels. Scale bar = 5 μm. Right, the percentage of TUG1 spots colocalizing with γ-H2AX relative to the total number of

TUG1 spots in each cell. **d** RIP assay using GFP antibody indicating the interaction between RPA32 or RPA70 and TUG1 after treatment with 10 μM CPT in HeLa cells expressing RPA32-GFP or RPA70-GFP. HPRT and IgG-bound RNA were taken as negative controls. Data are mean ± SD, *n* = 3. Two-sided *t*-test. **e** Left top, super-resolution images of HeLa cell transfected with catalytically inactive RNase H1 (RNH1^D210N^-GFP, green) co-stained with pRPA32 (white) and TUG1 smFISH (red). Cell was treated with DMSO or 10 μM CPT for 2 h. Left bottom panels are magnified regions. Scale bar = 5 μm. Right, the percentage of TUG1 spots colocalizing with RNH1^D210N^-GFP relative to the total number of TUG1 spots in each cell. For **b, c,** and **e**, median, upper and lower quartile range from 10 independent cells are indicated. Two-sided *t*-test. Three independent experiments were carried out with similar results (Supplementary Table 1) and a representative image is shown. Source data are provided as a Source Data file.

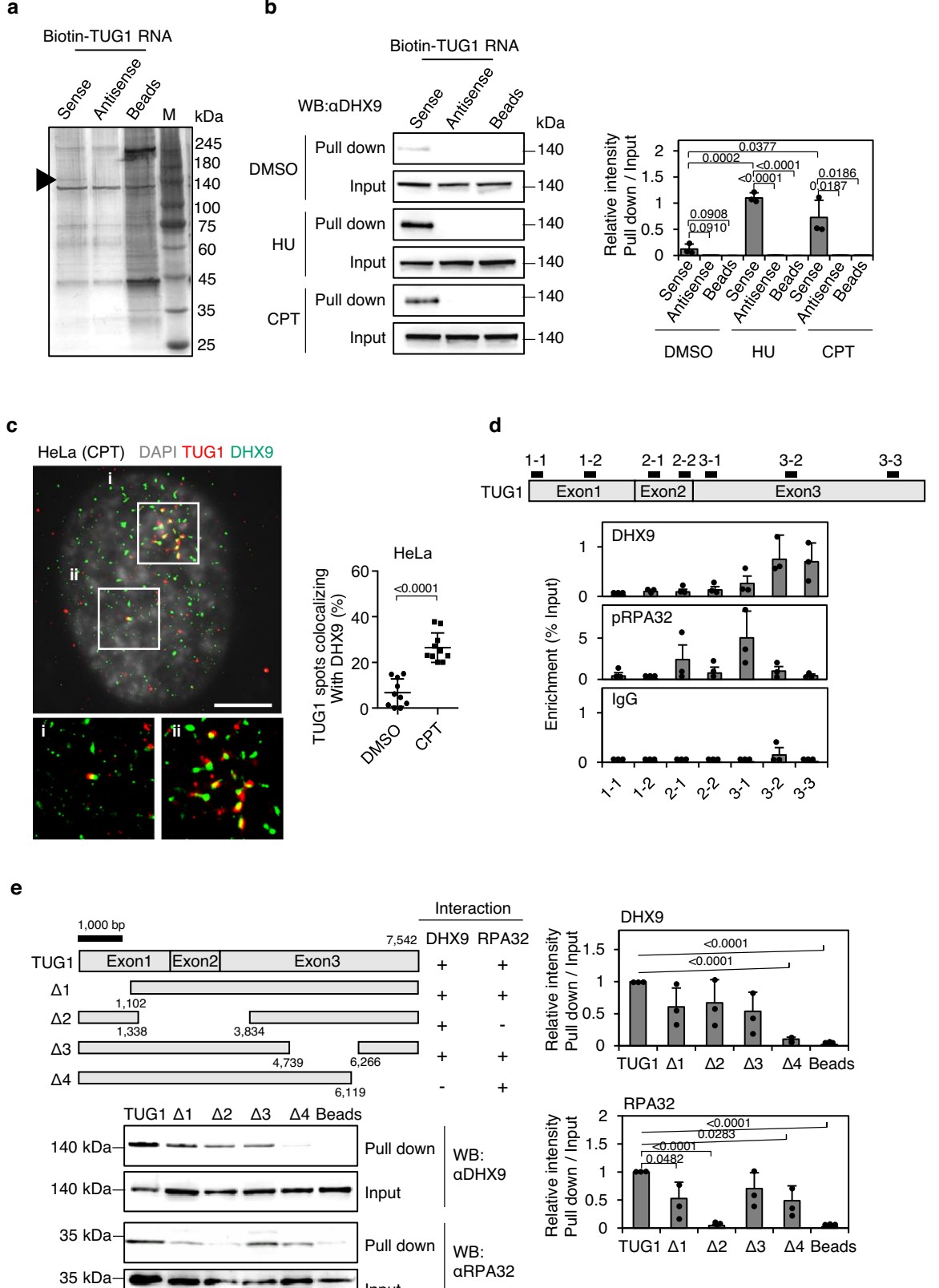

formation at the transcribed regions, proving the effects of transcription on R-loop formation in this system (Fig. 4f). The accumulation of R-loops was significantly reduced by tethering TUG1-PP7 to the LacO-repeat, without disturbing *CFP* expression (Fig. 4e, Supplementary Figs. 5b, 5c). In contrast, tethering the Δ4 deletion mutant of TUG1-PP7, which DHX9 cannot bind (Fig. 3d, e), did not suppress R-loop

formation (Fig. 4g). Furthermore, the Δ3 deletion mutant of TUG1-PP7, with which DHX9 still interacts in the CLIP assay (Fig. 4g), did suppress R-loop formation, albeit to a lesser extent than full-length TUG1. R-loop formation was also significantly suppressed by directly tethering DHX9 to the LacO-repeat (Supplementary Fig. 5d–f). However, tethering the DHX9 helicase-dead mutant (D511A, E512A)[40] did not suppress R-loop

**Fig. 3 | TUG1 interacts with DHX9. a** Silver-staining gel image of RNA pull-down experiment using Biotin-labeled TUG1 RNA and HeLa cells treated with 2 mM HU for 2 h. Lane M, marker proteins (sizes in kDa). The arrowhead indicates DHX9. Two independent experiments were carried out with similar results and a representative image is shown. **b** Left, Western blotting of DHX9 pulled down with TUG1. HeLa cells treated with DMSO (untreated control), 2 mM HU, or 10 μM CPT for 2 h were used. Right, bar graph generated by quantifying the Western blot. Data are means ± SD, *n* = 3. Two-sided *t*-test. **c** Top, super-resolution images of HeLa cells treated with 10 μM CPT for 2 h were co-stained with DAPI (white), DHX9 (green), and TUG1 smFISH (red). Bottom panels are magnified regions where TUG1 and DHX9 colocalize in the top panels. Scale bar = 5 μm. Right, the percentage of TUG1 spots colocalizing with DHX9 relative to total number of TUG1 spots in each cell. HeLa cells were treated with DMSO or 10 μM CPT for 2 h. Median, upper and lower quartile range from 10 independent cells are indicated. Two-sided *t*-test. Three independent experiments were carried out with similar results (Supplementary Table 1) and representative images are shown. **d** CLIP assay of DHX9 and pRPA32 performed in HeLa. Associated TUG1 RNA was quantified by qPCR using primers indicated in the upper TUG1 scheme. Data are presented as % input, mean ± SD, *n* = 3. **e** Left, Western blotting of DHX9 and RPA32 pulled down with TUG1-full length (TUG1) or TUG1-deletion RNA (Δ1-Δ4) indicated in the upper TUG1 scheme. See also Supplementary Fig. 3a. Right, bar graph generated by quantifying the Western blot. Signal intensities are normalized to the Input. Values are relative to TUG1-full length. Data are means ± SD, *n* = 3. Two-sided *t*-test. Source data are provided as a Source Data file.

formation (Supplementary Fig. 5f), indicating helicase-activity-dependency of R-loop resolution. Consistent with this, KD of DHX9 also increased the amount of RNA/DNA hybrids in this experimental setting (Fig. 4h). These data suggest that TUG1 bound by DHX9 resolves R-loops induced by transcription.

Note that tethering the Δ2 deletion mutant of TUG1-PP7, where RPA32 bound, suppressed R-loop formation to the same extent as full-length TUG1-PP7 in this assay (Fig. 4g). This suggests that direct interactions between RPA32 and TUG1 are not required for R-loop resolution, but may play another role, such as in recruiting TUG1 to R-loop regions.

## TUG1 is involved in the resolution of R-loops in (CA)n/(TG)n repeat-containing loci

To examine the distribution of R-loops specifically regulated by TUG1, DNA:RNA immunoprecipitation coupled with high-throughput DNA sequencing (DRIP-seq) analysis using S9.6 antibody was carried out after treatment with TUG1 ASO (Supplementary Fig. 6a, b). HeLa cells were also treated with CPT, which globally promotes R-loop accumulation[7,41,42]. Both TUG1 depletion and CPT treatment mainly increases DRIP-seq peaks (Fig. 5a). The total number of R-loop peaks that increased after TUG1 depletion was much smaller than after CPT treatment (528 versus 19,191 peaks, respectively, Fig. 5a, b). Among the 528 TUG1-sensitive regions, in which R-loops (i.e., read counts in DRIP-seq) were profoundly accumulated after TUG1 KD, 247 overlapped with CPT-sensitive regions (Fig. 5b). Interestingly, in the TUG1-sensitive regions, such as intronic regions of the *BCL2*, *PRS6KA2* and *C22orf34* genes, which harbor CA repeats, R-loops had already accumulated to a certain degree even in untreated cells. These were further increased by treatment with either TUG1 KD or CPT (whereby TUG1 KD > CPT), and co-treatment with TUG1 KD and CPT in HeLa cells (Fig. 5c, d, Supplementary Fig. 6c, d). We acknowledge that the levels of R-loop induced by co-treatment with TUG1 KD and CPT are lower than for CPT treatment alone in this set of experiments (Fig. 5c). This is probably because the combination of TUG1 KD and CPT treatment for 2 h suppressed total cellular metabolism including transcription and DNA replication, resulting in suppression of overall R-loop formation in a certain population of cells[42].

Although we did not see overlap between TUG1 and γ-H2AX two hours after CPT treatment (Fig. 2c), analysis of public ChIP-seq data[43] of untreated non-synchronized HeLa cells revealed that γ-H2AX tends to be enriched at locations around TUG1-sensitive regions (Fig. 5e). This suggests that the TUG1-sensitive regions are susceptible to R-loop accumulation and consequent DNA damage in cancer cells even without any treatment (i.e., in the unperturbed state). In contrast, de novo R-loop peaks were dominantly observed in CPT-sensitive regions, although mean levels of R-loop accumulation were lower than in the TUG1-sensitive regions (Fig. 5c).

We next investigated the characteristics of TUG1-sensitive R-loop-forming regions. TUG1-sensitive R-loop peaks were significantly longer than CPT-sensitive peaks (median length of 1021 bp and 773 bp, respectively. Supplementary Fig. 6e). Both TUG1-sensitive and CPT-sensitive peaks were primarily found in intergenic and intron regions (Supplementary Fig. 6f, Supplementary Table 4). Notably, about 80% of TUG1-sensitive regions contain simple repeats (Fig. 5f). Specifically, 63% of TUG1-sensitive regions were (CA)n and (TG)n repeat-containing sequences (Fig. 5d, g, Supplementary Fig. 6c, d, g). In contrast, only a minor fraction of CPT-sensitive peaks (<10%) arose at (CA)n and (TG)n repeat-containing sequences (Figs. 5f, g, Supplementary Fig. 6g). Validation analysis by DRIP-qPCR showed that TUG1 depletion by ASO increased R-loops at the TUG1-sensitive CA repeat regions (*BCL2*, *PRS6KA2* and *C22orf34*)(Supplementary Fig. 6h). Similarly, depletion of DHX9 also effectively increased R-loops in these regions (Supplementary Fig. 6i). In order to investigate whether TUG1 binds TUG1-sensitive regions, we further performed CARPID-based ChIP-qPCR and validated that TUG1 and proteins in its proximity were significantly enriched at those regions (Supplementary Fig. 6j). Consistent with this, the TUG1-sensitive CA repeat regions were also enriched with DHX9 (Supplementary Fig. 6k).

Interestingly, we found that continuous TUG1 depletion for more than a week enhanced microsatellite instability (MSI) in the three dinucleotide markers (D2S123, D5S346 and D17S250) of the Bethesda reference panel that contains CA repeats[44] in the DNA mismatch repair (MMR)-proficient cell lines (Supplementary Fig. 7a, b). The level of expression of the MMR genes *MLH1*, *MSH2*, *MSH6*, *PMS1* and *PMS2* was not decreased by TUG1 depletion (Supplementary Fig. 7b).

## TUG1 depletion results in marked DNA damage and apoptosis

As mentioned above, after four hours of TUG1 depletion, R-loops were substantially increased (Figs. 4b, 5a) and the speed of fork progression reduced (Fig. 4c, Supplementary Fig. 4d). Consequently, TUG1 depletion efficiently increased the levels of DNA damage marker γ-H2AX in six hours of treatment with TUG1 ASO in HeLa/Fucci2 cells (Fig. 6a). After treatment with TUG1 ASO for 24 h, FCM showed significant decrease in DNA replication rate (i.e. EdU incorporation, Supplementary Fig. 4f) and increased numbers of γ-H2AX-positive cells in the S phase of HeLa/Fucci2 and other cancer cell lines (Fig. 6b, Supplementary Fig. 8a, b).

The levels of the p-ATR, p-CHK1, p-ATM, and p-CHK2 proteins, which are key DNA damage-response checkpoint proteins, were elevated in cells treated with TUG1 ASO relative to control cells (Supplementary Fig. 8c). DNA fragmentation (i.e. DSBs) assessed by the neutral comet assay was significantly increased in TUG1-depleted cells (Fig. 6c, Supplementary Fig. 8d). DNA fragmentation appeared to be mostly induced by RS but not by apoptotic fragmentation, because no significant difference in the tail moment between cells treated with or without caspase inhibitor (Z-VAD-FMK)[45] was observed (Supplementary Fig. 8e). These results indicate that the accumulated RS caused by 24-h TUG1 depletion was processed as prominent DNA damage.

Consequently, depletion of TUG1 inhibited cell growth (Fig. 6d) and increased apoptosis, as evidenced by the Annexin V assay in cancer cells after treatment with TUG1 ASO for 48 h (Fig. 6e). In contrast, depletion of TUG1 minimally affected both cell proliferation and apoptosis in normal cells, supporting the idea that TUG1 is a pivotal

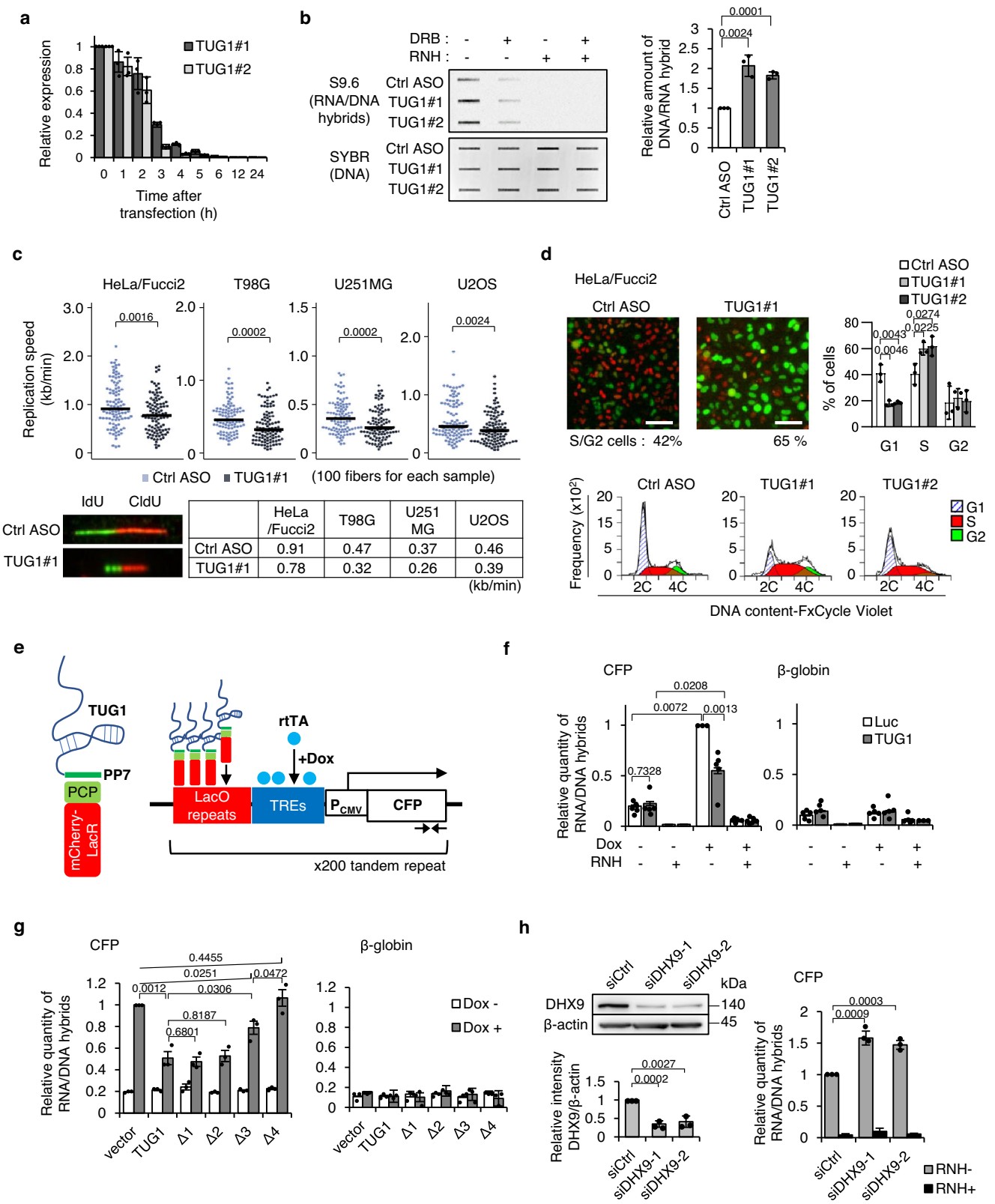

molecule to regulate R-loop resolution and maintain cancer cell proliferation, as we hypothesized (Fig. 6f, g). Note that co-treatment of TUG1 ASO and three DNA-damaging agents with distinct modes-of-action (cisplatin, Temozolomide (TMZ) and CPT) showed synergistic induction of apoptosis and suppression of the growth of cancer cells including glioblastoma (Supplementary Fig. 9a–d). Combination index estimates[46] revealed that TUG1 depletion synergistically enhanced the

effects of CPT, TMZ, and cisplatin (combination index, 0.7, 0.8, and 0.7, respectively, Supplementary Fig. 9a–d). Taken together, we conclude that replication delay is mainly caused by slowed replication fork progression early in the response to TUG1 depletion (i.e. four hours of TUG1 KD), while at a later time point (i.e. 24 to 48 h), a proportion of cells arrest or die due to DNA damage, resulting in suppression of cell growth.

**Fig. 4 | TUG1 resolves R-loops via DHX9 activity. a** KD efficiency by ASOs analyzed by RT-qPCR in HeLa/Fucci2 cells at indicated time after transfection. Mean ± SD, $n = 3$. **b** Left, Slot blot with S9.6 antibody. HeLa/Fucci2 cells transfected with ASOs for 4 h were treated with 100 μM DRB for the last 40 min to show transcriptional source of the R-loops. Genomic DNA was treated with RNase H (RNH) in vitro before blotting. SYBR gold staining is a loading control. Right, quantification of S9.6 signal. Mean ± SD, $n = 3$. Two-sided $t$-test. **c** Top, DNA fiber assay in four cell lines transfected with ASOs for 4 h. Black bar, median value. Mann–Whitney U test. Bottom left, representative fiber images. Bottom right, median replication speeds. The experiment was repeated three times (Supplementary Table 3). **d** Top left, HeLa/Fucci2 cells transfected with ASOs for 24 h. S/G2 cells are shown in green. Scale bar = 50 μm. Top right, cell-cycle analysis of cells treated with ASOs for 24 h by FCM. Mean ± SD, $n = 3$. Two-sided $t$-test. Bottom, representative profiles of cell-cycle distribution after 24 h of TUG1 KD. **e** Schematic presentation of tethering of

TUG1 on LacO-repeat regions by LacO-LacR and PP7-PCP system in U2OS 2-6-3/TA cells combined with Doxycycline (Dox)-inducible transcriptional activation. Arrows indicate primer positions for DRIP-qPCR. **f** Quantification of RNA/DNA hybrids accumulated in transcribed CFP and β-globin gene (control) by DRIP-qPCR assay (see also Supplementary Fig. 5a). **g** As in **f**, relative amount of RNA/DNA hybrids in U2OS 2-6-3/TA cells expressing deletion mutants of TUG1 compared with those in the cells without TUG1 expression (vector) under Dox induction. For **g** and **f**, Mean ± SE, $n = 5$. Two-sided $t$-test. **h** Top left, levels of DHX9 in U2OS 2-6-3/TA cells transfected with siCtrl, or siDHX9 for 48 h. Bottom left, bar graph generated by quantifying the Western blot. Data are means ± SD, $n = 3$. Two-sided $t$-test. Right, as in **f**, relative amount of RNA/DNA hybrids in U2OS 2-6-3/TA cells transfected with siDHX9. Mean ± SE, $n = 3$. Two-sided $t$-test. Source data are provided as a Source Data file.

## Combination therapy with antiTUG1-DDS and TMZ suppresses tumor growth

Finally, we addressed whether targeting TUG1 is an effective therapeutic approach to enhance the efficacy of chemotherapies. TMZ is the first-line drug to treat glioblastoma, however, its efficacy and clinical application are limited[47]. TUG1 colocalizes with pRPA32 at R-loop (Supplementary Fig. 10a, b) and DHX9 (Supplementary Fig. 10c) in the glioblastoma cell line LN229. Treatment of LN229 with TUG1 ASO and TMZ resulted in the accumulation of RNA/DNA hybrids; moreover, treatment with both TUG1 ASO and TMZ synergistically induced apoptosis (Fig. 7a, b). Next, we exploited the LN229 xenograft mouse model to investigate the efficacy of intravenous treatment with TUG1 ASO coupled with a tumor-specific drug delivery system (antiTUG1-DDS) in vivo[48] (Fig. 7c). Combining antiTUG1-DDS with TMZ most effectively suppressed tumor growth, compared to either TMZ or antiTUG1-DDS alone (Fig. 7d, e, Supplementary Fig. 10d). This combination treatment markedly prolonged the overall survival, more than antiTUG1-DDS or TMZ alone (Fig. 7f). Importantly, no apparent adverse effects were observed in any of the treated animals. Reduced expression of TUG1 and increased levels of R-loops were consistently observed in the antiTUG1-DDS treated tumors (Fig. 7g, h, Supplementary Fig. 10e).

The DNA methylation status in the O-6-methylguanine-DNA methyltransferase (*MGMT*) gene promoter is a prognostic factor in GBM in that a high level of DNA methylation is correlated with TMZ sensitivity[49]. Because LN229 has a high level of DNA methylation in the promoter of *MGMT* (82.7% by pyrosequencing analysis), we examined the effect of TUG1 ASO and TMZ in another glioblastoma cell line U251MG, which has a lower level (23.2%) of DNA methylation in the *MGMT* promoter (Supplementary Fig. 10f). Also for this cell line, combination therapy resulted in significantly greater suppression of tumor growth than either TUG1 ASO or TMZ alone (Supplementary Fig. 10g–i).

## Discussion

Although escaping critical checkpoints enables cell proliferation during tumorigenesis, a checkpoint-coupled repair system is paradoxically required to protect transformed cells from excessive RS and replication catastrophe[50]. In the current study, we found that lncRNA TUG1 is a key molecule that is rapidly upregulated in cancer cells via activation of the ATR-CHK1 signaling pathway and is engaged in resolving overabundant R-loops at certain loci, particularly at CA repeat microsatellite regions (Supplementary Fig. 11). The basal expression levels of TUG1 were higher in different types of cancer cells than normal cells, which may reflect the functional roles of TUG1 on intrinsic R-loop resolution.

Studies have shown multiple functions of TUG1, including roles in retinal development, cell-cycle regulation, and tumorigenesis[17,24,51–53]. Our findings provide additional insights into the biological importance of TUG1. Accumulating studies are demonstrating the involvement of

lncRNAs in the regulation of the cell cycle and maintenance of genome stability[54,55]. However, no study to date has shown global regulation of R-loop formation by lncRNAs in cancer cells. Because RPA and DHX9 are present in cancer cells throughout the cell cycle, expression of TUG1 via the ATR-CHK1 pathway may foster immediate interactions between these molecules in order to prevent overabundant R-loop accumulation specifically at S phase.

TUG1 upregulation was most likely not due to direct regulation by *TP53*. Our previous studies showed that MYC drives the expression of TUG1 in cancer stem cells[24]. In the current study, in addition to fundamental expression control of TUG1 by MYC (Supplementary Fig. 1h), we found the ATR-CHK1-E2F pathway directly upregulated TUG1 expression in response to HU and CPT exposure (i.e. two hours of treatment), which induced more abundant RS and R-loop, in cancer cells. Bertoli et al. reported that E2F-mediated regulation via CHK1-dependent phosphorylation is a key mechanism underlying tolerance to RS[23,56]. Sustained E2F1 transcription and inactivation of E2F6 in S phase upregulates many proteins associated with the RS checkpoint response, such as those stabilizing ongoing replication forks. Among such E2F target molecules, TUG1 appears to be an important early-response molecule affecting R-loops immediately after its transcription, acting before genome integrity-associated proteins are synthesized.

Substantial (>90%) depletion of TUG1 expression by ASO was achieved even within two to four hours after transfection in the four cell lines examined. R-loop accumulation and reduction of fork speed were observed within four hours of treatment with ASO (Supplementary Fig. 11). Thus, our timecourse experiment of TUG1 depletion apparently excluded the possibilities of other TUG1 functions involved in the regulation of R-loop formation, such as altered protein expression resulting from silencing through PRC2 interaction, production of functional peptides, or interacting with microRNAs[17,24] in the current study[17].

One study has shown that head-on oriented transcription-replication conflicts promote R-loop formation and ATR activation[57]. During this process, ATR responds to stretches of RPA-coated ssDNAs at stalled replication forks[58]. Another study showed that RPA can recognize R-loops independently of DNA replication and DSBs, and then suppress R-loop accumulation via its interaction with RNase H1[8]. Additionally, RPA-coated ssDNAs at centromeres trigger R-loop-dependent ATR activation during mitosis to promote faithful chromosome segregation[59]. In the current study, we found clear evidence of colocalization of TUG1 and pRPA32 at R-loop loci using super-resolution single-molecule fluorescence microscopy. In addition, a certain amount of TUG1 is colocalized with PCNA, indicating that R-loops to which TUG1 binds are formed at replication forks. This is consistent with the finding that depletion of TUG1 resulted in R-loop accumulation concurrent with reduced fork speed.

Using CLIP and other biochemical analyzes, we further found that TUG1 directly and functionally interacted with DHX9 and RPA32. These

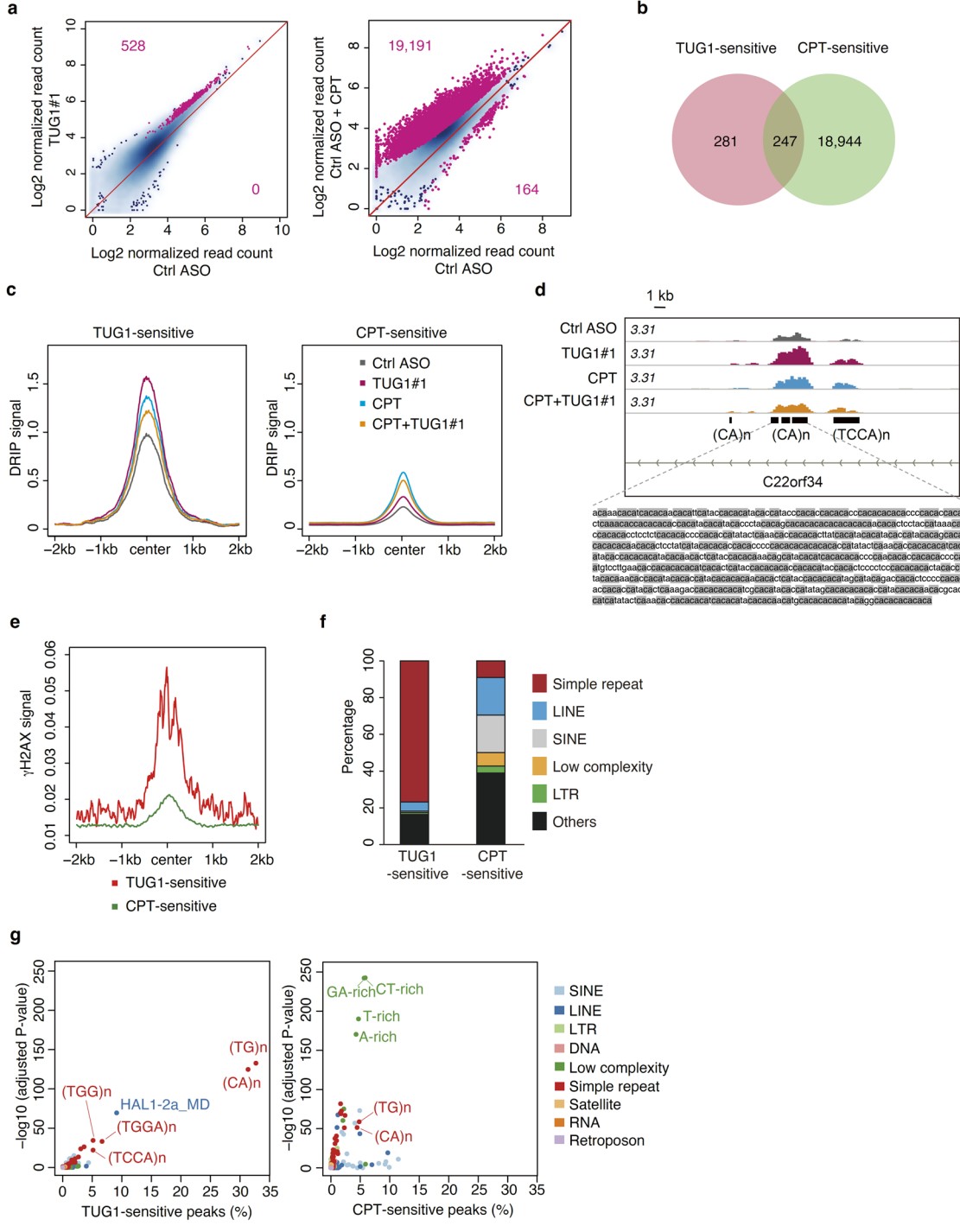

**Fig. 5 | TUG1 resolves (CA)n/(TG)n containing R-loops. a** Scatter plots comparing the mean log2 concentration of DRIP-seq signals. HeLa/Fucci2 cells transfected with Ctrl ASO or TUG1#1 for 4 h were treated with or without 10 μM CPT for the last 2 h. TUG1#1-treated cells (left) and CPT-treated cells (right) against control samples. Peaks with false discovery rate (FDR) < 0.1 are colored magenta, and the number above and below the diagonal line represents up- and down-regulated peaks after the treatment, respectively. DRIP-seq signals were normalized based on library size. **b** Venn diagram showing the overlap between TUG1-sensitive and CPT-sensitive peaks. **c** Metaplot of mean input-subtracted DRIP-seq signals in 4 kb window around the TUG1-sensitive (left) and CPT-sensitive (right) peak centers. **d** A snapshot of representative locus of DRIP-seq peaks enhanced by TUG1 KD in chr22:49,973,028-49,975,721 (C22orf34). (CA)n repeat-containing sequence is

indicated below with CA dinucleotide colored gray. **e** Metaplot of γ-H2AX accumulation in 4 kb window around the TUG1-sensitive (red) and CPT-sensitive (green) peak centers. **f** Genomic annotation of peaks differentially altered by TUG1 KD or CPT treatment, defined by homer. Percentages of repeat types (>1%) are detailed in the plot. **g** Simple repeat enrichment analysis in TUG1-sensitive (left) or CPT-sensitive (right) peaks. The x axis shows the fraction of TUG1- or CPT-sensitive peaks overlapping with the indicated repeat types and sequences in RepeatMasker. The y-axis shows negative log10 P-values of enrichment or depletion of indicated repeat types and sequences above background. P-values were estimated by comparison with GC%-matched background regions for TUG1- or CPT-sensitive peaks using two-sided Fisher's exact-test and adjusted using the Benjamini–Hochberg method.

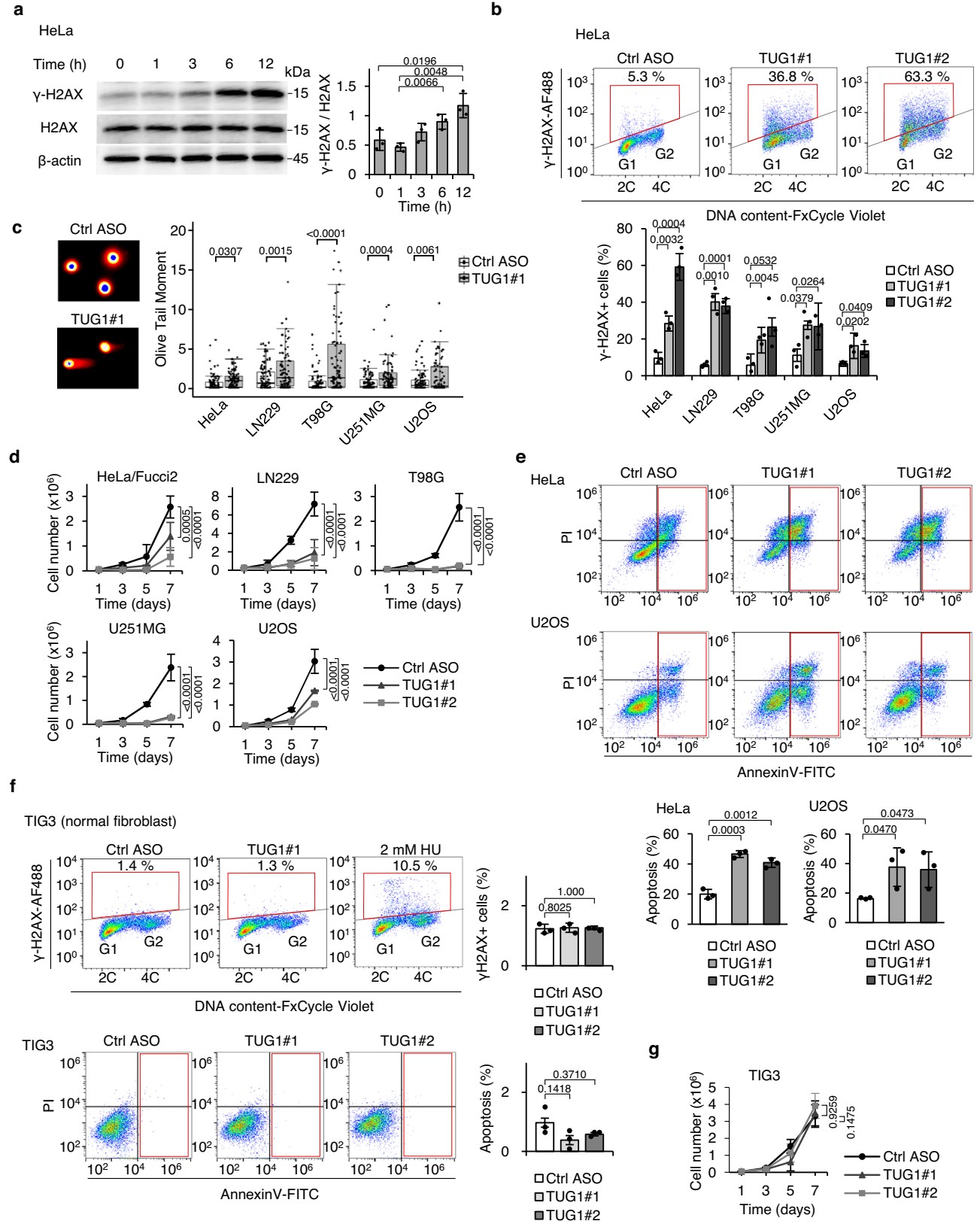

data indicate that TUG1 suppresses R-loops together with DHX9 and RPA32 that can recognize ssDNA at or near R-loop regions. Notably, a previous study had shown that RNase H1 bound RPA70, but not RPA32, because this interaction was not prevented by RNase A treatment, suggesting that RNAs are not involved in RPA70-RNase H1 interactions[8]. In addition to RNase H enzymes, different helicases including DHX9, SETX, and DDX5 have also been shown to unwind RNA/DNA hybrids and resolve R-loops[60]. Recently, RNA/DNA proximity proteomics technology that can map the R-loop proximal proteins using quantitative mass spectrometry was used to identify different functional cellular proteins in addition to helicases in R-loop regulation; these results suggested that R-loops formed during

**Fig. 6 | TUG1 depletion results in marked DNA damage and apoptosis. a** Left, Western blot showing the level of γ-H2AX in cells transfected with TUG1#1 for the indicated time. Right, the relative intensities of γ-H2AX normalized by H2AX. Mean ± SD. $n = 3$. Two-sided $t$-test. **b** Top, representative FCM profiles of HeLa/Fucci2 cells after 24 h of TUG1 KD. Percentage of γ-H2AX-positive cells (gated cells) are indicated. 2 C and 4 C indicate non-replicated and replicated genomes, respectively. Bottom, Percentage of γ-H2AX-positive cells after 24 h of TUG1 KD in five cell lines. Mean ± SD. $n = 3$. Two-sided $t$-test. **c** Detection of DSBs by a neutral comet assay. Left, representative images of comet tails in Ctrl ASO or TUG1#1 transfected cells. Right, quantification of Olivetail moment. 100 cells per group were examined. One-way ANOVA. The experiments were carried out in duplicate with similar results. **d** Cell growth after transfection with Ctrl ASO, TUG1#1 or TUG1#2 for the indicated times. Mean ± SD. $n = 3$. One-way ANOVA and Tukey's

multiple comparison tests. **e** Top, Annexin V/propidium iodide (PI) staining of HeLa and U2OS cells transfected with Ctrl ASO, TUG1#1 or TUG1#2 for 48 h. Annexin V-positive (apoptotic) cells are gated. Bottom, percentages of Annexin V-positive cells. Mean ± SD. $n = 3$. Two-sided $t$-test. **f** Top left, representative FCM profiles of TIG3 normal fibroblast cells after 24 h of TUG1 KD or treatment of HU for 24 h. Percentage of γ-H2AX-positive cells (gated cells) are indicated. Top right, bar graph shows the percentage of γ-H2AX-positive cells after 24 h of TUG1 KD in TIG3 cells. Mean ± SD. $n = 3$. Two-sided $t$-test. Bottom left, Annexin V/PI staining of TIG3 cells transfected with Ctrl ASO, TUG1#1 or TUG1#2 for 48 h. Annexin V-positive (apoptotic) cells are gated. Bottom right, bar graph shows the percentages of Annexin V-positive cells. Mean ± SD. $n = 3$. Two-sided $t$-test. **g** Cell growth of TIG3 after transfection with Ctrl ASO, TUG1#1 or TUG1#2 for the indicated times. Mean ± SD. $n = 3$. One-way ANOVA. Source data are provided as a Source Data file.

physiological or pathological process may be controlled by different protein complexes. Interestingly RPA70 and RPA32 were also identified as R-loop proximal proteins in an unbiased assay[61]. Taken together, it appears that the TUG1-RPA-DHX9 interaction is an indispensable lncRNA-mediated mechanism for regulating R-loops, in addition to previously reported mechanisms[60].

After CPT treatment, R-loops colocalized with both fully spliced and intron-retained TUG1 transcripts (Supplementary Fig. 2d). These data suggest that exon regions may be sufficient to resolve R-loops. However, it is unclear whether these two variants function in any particularly favored manner. Further work is needed to define the functional difference between fully spliced and intron-retained TUG1 transcripts[35].

We found that TUG1 removes pathological R-loops susceptible to DNA damage in cancer cells. Depletion of TUG1 increased R-loop accumulation especially at (CA)n/(TG)n microsatellite repeat regions. This finding indicates that TUG1 molecules appear to be involved in resolution of R-loops in a genomic location-specific manner; thus, TUG1 is one of the important regulatory molecules for locus-specific R-loop resolution mechanisms. Studies showed that pathological R-loops formed at CAG microsatellite regions cause instability at expandable triplet repeat sequences[62,63], which is associated with onset of neurodegenerative diseases via alteration of gene expression[64,65]. During the CAG expansion process, MutLγ (i.e. MLH1, MLH3) nuclease activity causes both R-loop-induced CAG fragility and contractions by nicking R-loop-induced structures[66,67]. CA repeats are another type of microsatellite repeats that are distributed throughout the genome and affect transcription of nearby genes[68]. For example, the MeCP2 protein is a microsatellite DNA-binding protein that targets the CA-rich strand and controls nucleosome-free genomic regions[69]. MSI, mainly caused by deficiency in MMR system, is a form of genomic instability characteristic for cancer cells[70]. We found that TUG1 depletion caused accumulation of R-loops at or near CA microsatellite repeat regions together with induction of MSI in cancer cells regardless of MMR status. In clear contrast, CPT treatment, which promotes global R-loop accumulation[21], caused the accumulation of a larger number of R-loops with R-loop profiles different from those resulting from TUG1 depletion. Although the underlying mechanisms responsible for the preferential engagement of TUG1-RPA-DHX9 for R-loop resolution at CA microsatellite regions remain to be elucidated, our data indicate that TUG1 is involved in resolution of R-loops at specific loci in cancer cells. Disruption of these is associated with induction of MSI probably due to continuous R-loop-induced DNA damage and repair processes[66].

DHX9 unwinding activity proceeds in the $3' \rightarrow 5'$ direction and requires a 3'-overhang[71–73]. Based on this finding, DHX9 does not appear to directly unwind RNA-DNA in canonical R-loops with a 5' ssRNA tail. Instead, DHX9 plays a key initial role in R-loop resolution by destabilizing the formed R-loops. The helicase activity of DHX9 may act to resolve R-loops with secondary structures in ssDNA, such as at G-quadruplexes[73] and long microsatellite sequences[74,75]. Alternatively, DHX9 may bind to TUG1 to resolve R-loops with 3' ssRNA tails, which

are formed by RNA polymerase II backtracking[76]. In particular, long microsatellite sequences induce stable R-loops and may represent prime sites for pausing and backtracking of RNA polymerase II[77]. Further studies will clarify whether DHX9 can resolve R-loops with 3' ssRNA formed by RNA polymerase II backtracking.

There is a growing interest in targeting oncogene-induced replication stress as a novel approach to cancer therapy[78]. We have reported the development of a TUG1-ASO DDS using cyclic peptide-conjugated polymeric micelles[24] or Y-shaped block catiomers[48], which can be used intravenously. Such cancer-specific DDSs may become an effective therapeutic option for treatment of refractory cancer. As mentioned above, pathogenic R-loop formation is strictly regulated at an appropriate level in normal cells[60]. Given the fact that substantial RS and unscheduled R-loops are not observed in normal cells but are common features of most precancerous as well as cancerous cells[1], targeting TUG1 could represent a cancer-specific therapeutic strategy. Furthermore, combining TUG1 ASO treatment with the alkylating agent TMZ, a first-line treatment for the most common lethal brain tumor, glioblastoma[47], synergistically enhances anti-tumor effects on glioblastoma cells. Although further investigation is required, TUG1 ASO may be a powerful strategy for targeting this dire tumor entity, particularly because of its synergetic effects with TMZ or Irinotecan, an analog of CPT also clinically used for glioblastoma treatment. Interestingly, we found that TUG1 ASO treatment consistently induced MSI in dinucleotide markers of the Bethesda reference panel (D2S123, D5S346 and D17S250), indicating that high levels of microsatellite instability (MSI-H) were induced, at least in part, in those cells[44]. A recent study revealed that replication stress-associated DSBs induce MSI in MMR-deficient cells[79]. We found here that TUG1 depletion induced MSI even in MMR-proficient cells. Because immune checkpoint inhibitors (ICI) are used for the treatment of heavily mutated cancers with MSI-H[80], it might be an attractive treatment strategy to combine them with TUG1 ASO, even for cancers where the MSI-H molecular phenotype is uncommon.

In conclusion, our data reveal an important function of TUG1 as an indispensable RNA molecule controlling R-loops in proliferating cancer cells. We provide a new paradigm whereby targeting TUG1 by ASO, coupled with DNA-damaging agents, may be an effective novel strategy for the treatment of cancers, particularly those with a high RS and R-loop burden.

## Methods

### Cell culture and transfection

HeLa/Fucci2 cells were obtained from the RIKEN Cell Bank, Japan. HeLa/Fucci2 utilize the fluorescent ubiquitination-based cell-cycle indicator (Fucci) system, which visualizes cell-cycle progression in live cells; G1 cells are mCherry-positive (shown in red), and S/G2 cells are mVenus-positive (shown in green)[19]. HeLa/Fucci2, HeLa (RIKEN Cell Bank), LN229 (The American Type Culture Collection (ATCC)), U251MG (JCRB Cell Bank, Japan), U2OS (ATCC), TIG3 (RIKEN Cell Bank), and HEK293T (RIKEN Cell Bank) cell lines were cultured in Dulbecco's

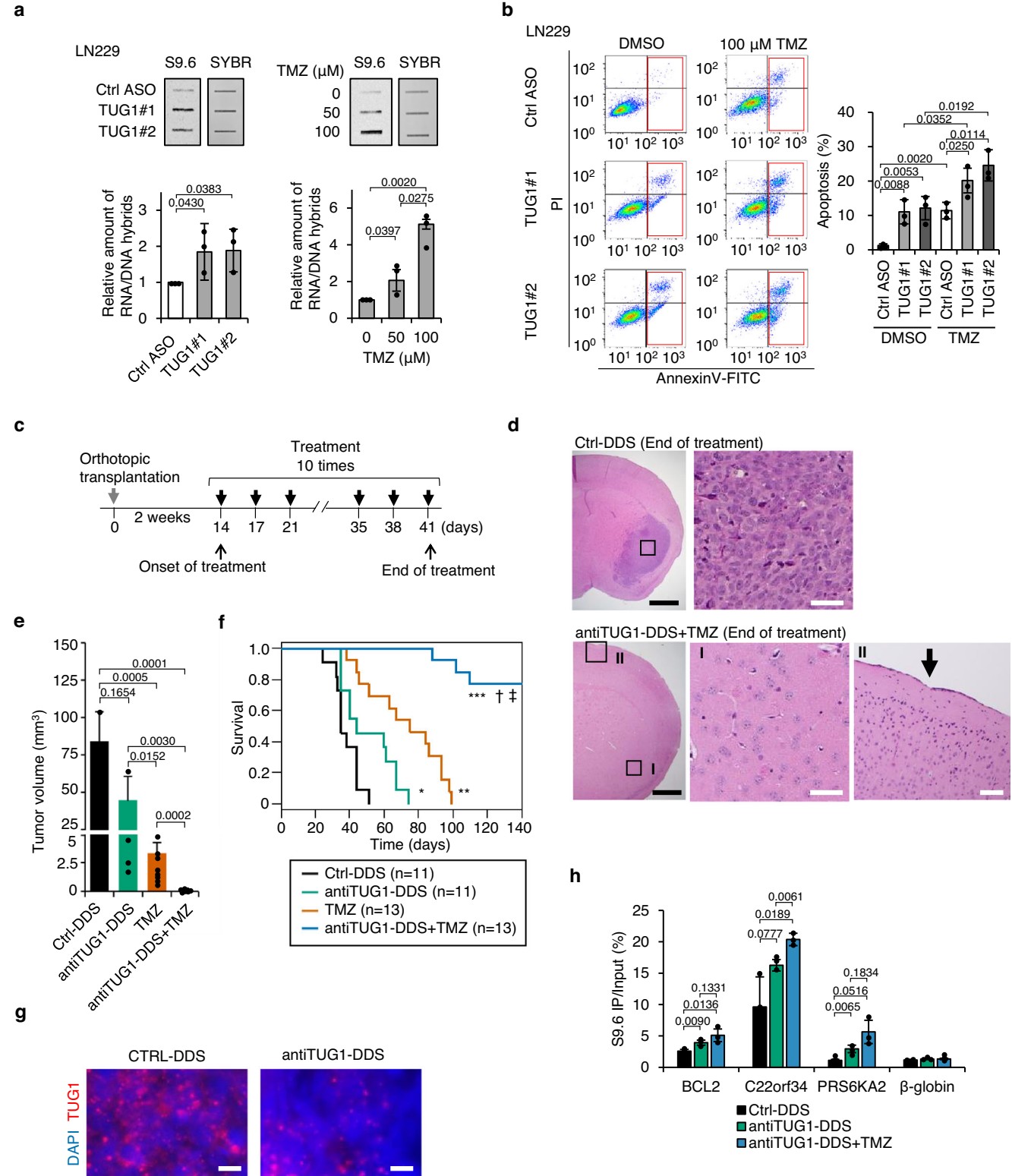

modified Eagle's medium (DMEM) containing 10% fetal bovine serum (FBS) and 1x antibiotic-antimycotic (Anti-anti, Gibco, Thermo Fisher Scientific, MA, USA). T98G (RIKEN Cell Bank) cells were cultured in RPMI1640 containing 10% FBS with 1x Anti-anti. All the cell lines used in this study are microsatellite stable (MSS). U2OS 2-6-3 (provided by David L. Spector at Cold Spring Harbor Laboratory)[39] cell line were cultured in DMEM containing 10% FBS and 1x Anti-anti supplemented with 1 μg/ml puromycin and 12.5 μg/ml hygromycin, respectively. Plasmid transfections and gene KD with ASO or siRNA were carried out

with Lipofectamine 3000 (Thermo Fisher Scientific), ViaFect Transfection Reagent (Promega, WI, USA), or ScreenFect A Plus (FUJIFILM Wako Pure Chemical, Japan) according to manufacturer's instructions. A list of the drugs, ASOs, siRNAs, and plasmids used is provided in Supplementary Data 2. For drug sensitivity assays and synergism analysis, cells were seeded at $2 \times 10^3$ cells/well into 96-well plates and treated with different concentrations of chemical reagents for 48 h. Cell viability was assessed using Cell Count Reagent SF (Nacalai Tesque, Japan). The assay was carried out in triplicate. Drug sensitivity was

**Fig. 7 | Combination therapy with antiTUG1-DDS and TMZ suppresses tumor growth in a glioblastoma xenograft mouse model. a** Top, RNA/DNA hybrid slot blot with S9.6 antibody. LN229 cells were transfected with Ctrl ASO, TUG1#1 or TUG1#2 for 4 h, or TMZ for 1.5 h at the indicated concentrations. DNA stained by SYBR gold is shown as a loading control. Bottom, bar graphs indicate quantification of S9.6 signals. Values are normalized to control and represent the mean ± SD, $n = 3$. Two-sided $t$-test. **b** Left, Annexin V/ propidium iodide (PI) staining of LN229 cells transfected with Ctrl ASO, TUG1#1 or TUG1#2 in combination with TMZ for 4 days. Annexin V-positive (apoptotic) cells are gated. Right, bar graph shows the percentages of Annexin V-positive cells. Mean ± SD. $n = 3$. Two-sided $t$-test. **c** Schematic diagram showing the treatment protocol for xenograft mouse models of LN229. **d** Representative HE-stained brain sections. Arrow shows the location of the puncture site. Boxed regions in the left panels are magnified in the right panels.

Black and white scale bars are 10 mm and 10 µm, respectively. **e** Tumor volumes at 35 days after transplantation. CTRL-DDS ($n = 11$), antiTUG1-DDS ($n = 13$), TMZ ($n = 15$), antiTUG1-DDS + TMZ ($n = 19$). Mean ± SE. Two-sided $t$-test. **f** Kaplan–Meier survival curves of mice treated as indicated. Statistical analysis was performed using the Log-rank test. *$P = 0.0185$ antiTUG1-DDS compared to CTRL-DDS, **$P = 0.0045$ TMZ compared to antiTUG1-DDS, ***$P = < 0.0001$ antiTUG1-DDS + TMZ compared to CTRL-DDS, †$P < 0.001$ antiTUG1-DDS + TMZ compared to antiTUG1-DDS, ‡$P < 0.001$ antiTUG1-DDS + TMZ compared to TMZ. **g** Representative smFISH image of TUG1 in CTRL-DDS and antiTUG1-DDS-treated tumors. Nuclei are stained with DAPI. Scale bars = 10 µm. **h** DRIP-qPCR of TUG1-sensitive loci in genomic DNA derived from human tumor xenografts after 1 week of treatment. Data are mean ± SD, $n = 3$. Two-sided $t$-test. Source data are provided as a Source Data file.

determined by half maximal inhibitory concentration (IC50) values using GraphPad Prism 8 (GraphPad Software, San Diego, CA, USA). To address if the synergistic or additive effects of the two treatments are obtained, combination index (CI) values were calculated using CompuSyn 1.0 (ComboSyn, Paramus, NJ, USA).

### Generation of stable cell lines
Cell lines with stable overexpression of target genes were generated using Lipofectamine 3000, according to the manufacturer's protocol. The HeLa and U2OS-derived cell lines that express GFP-tagged RPA70, HeLa-RPA70-GFP, or U2OS-RPA70-GFP, were established by selection with Blasticidin-S (2 ug/mL) following the transduction of pEF-BOS-RPA70-EGFP, in which RPA70 cDNA was replaced by the *H2B* gene on the N-terminus of EGFP in pEF-BOS-H2B-EGFP[81]. For cell lines with GFP-tagged RPA32, HeLa-RPA32-GFP, or U2OS-RPA32-GFP, G418 (100 ug/mL) was used for selection following the transduction of pEGFP-RPA32, in which RPA32 cDNA was fused to the C-terminus of EGFP of pEGFP-C1 (Takara Bio USA, CA, USA). U2OS 2-6-3 expressing rtTA, U2OS 2-6-3/TA, was established by transduction of the rtTA-expressing vector (pDisplay-rtTA). A list of the plasmids used is provided in Supplementary Data 2.

### Global analysis of lncRNAs
HeLa/Fucci2 cells were synchronized by mitotic shake off. After 13 h, S phase cells were treated with either DMSO, 2 mM HU, or 10 µM CPT for 2 h and examined. All experiments were performed in duplicate. Subcellular fractionation was conducted according to Pandya-Jones and Black[82]. RNA was extracted from the chromatin fraction using TRIzol reagent (Thermo Fisher Scientific). The RNA was amplified into cRNA and labeled according to the Agilent One-Color Microarray-Based Gene Expression Analysis protocol (Agilent Technologies, CA, USA). Labeled samples were purified with the RNeasy Kit (Qiagen, Germany) and hybridized to SurePrint G3 Human Gene Expression 8x60K v3 array slides (G4851C, Agilent Technologies) at 65 °C. The arrays were scanned using an Agilent Microarray Scanner (G2565BA, Agilent Technologies). The scanned images were analyzed using the Feature Extraction software, version 12.0 (Agilent Technologies), with background correction. The data analysis was performed with GeneSpring GX, version 7.3.1 (Agilent Technologies). LncRNA annotation is based on the Gencode v38 (GRCh38.p13).

### EdU incorporation assay
For labeling of newly synthesized DNA by EdU incorporation, cells were incubated with 100 µM EdU for 1 h and then processed using Click-iT™ Plus EdU Alexa Fluor™ 647 Imaging Kits (Thermo Fisher Scientific), according to the manufacturer's instructions.

### RT-qPCR
Total RNA was extracted using Trizol reagent (Thermo Fisher Scientific) followed by a reverse transcription using PrimeScript RT reagent Kit (Takara Bio). qPCR was conducted by THUNDERBIRD SYBR qPCR

Mix (TOYOBO, Japan)[24]. qPCR data was acquired using StepOne Software 2.3 (Thermo Fisher Scientific). The relative expression levels of target genes were determined using *GAPDH* as an internal control. Oligonucleotide primers are provided in Supplementary Data 2.

### smFISH
A smFISH experiment was performed using ViewRNA Cell Plus Assay Kit (Thermo Fisher Scientific) according to the manufacturer's instructions. $8 \times 10^3$ HeLa cells were cultured in 96-well half-area film bottom microplates (Corning, NY, USA, Cat. #4680) for 24 h before assay. The cells were imaged with a 40× objective lens using an Arrayscan VTI Microscope (Cellomics, MD, USA) coupled with the automated image analysis software HCS Studio Cellomics Scan Version 6.6.0 (Thermo Fisher Scientific). Image acquisition involved identification of 4′,6-diamidino-2-phenylindole (DAPI) stained cell nuclei as primary objects, followed by the application of a ring mask around the primary objects to identify a cytoplasmic area as secondary objects. Cells in the S/G2 and G1 phases were distinguished by a fluorescent ubiquitination-based cell-cycle indicator of HeLa/fucci2 cells. Probe sets used are shown in Supplementary Data 2.

### Western blotting
Target proteins were separated by SDS-PAGE and visualized on PVDF transfer membranes by specific antibodies[24]. Anti-α-Tubulin (ab64503, Abcam, 1:1000), anti-snRNP70 (sc-390899, Santa Cruz Biotechnology, CA, USA, 1:100), anti-Histone H3 (Cell signaling Technology, 4499, 1:2000), anti-phospho-Chk1 (Ser345) (Cell signaling Technology, 2341, 1:500), anti-Chk1 (sc-8408, Santa Cruz Biotechnology, 1:500), anti-phospho ATR (Thr1989) (58014 S, Cell signaling Technology, 1:500), anti-ATR (2790 S, Cell signaling Technology, 1:1000), anti-GAPDH (2118, Cell signaling Technology, 1:1000), anti-E2F1 (ab179445, Abcam, 1:1000), anti-E2F6 (ab53061, Abcam, 1:1000), anti-β-Actin (3700, Cell signaling Technology, 1:1000), anti-RNA Helicase A (DHX9) (ab26271, Abcam, 1:1000), anti-RPA32 (A300-244A-M, BETHYL, TX, USA, 1:1000), anti-RPA70 (NA13, Calbiochem, CA, USA, 1:100), anti-HA-tag (561, MBL, 1:1000), anti-phospho-Histone H2A.X (Ser139) (ab2893, Abcam, 1:1000), anti-phospho-Histone H2A.X (Ser139) (ab81299, Abcam, 1:5000), anti-phospho RPA32 (Ser33) (A300-246A-M, BETHYL, 1:1000), anti-phospho Chk2 (Thr68) (2197 S, Cell signaling Technology, 1:1000), anti-Chk2 (6334 S, Cell signaling Technology, 1:1000), anti-phospho ATM (Ser1981) (5883 S, Cell signaling Technology, 1:1000), and anti-ATM (ab201022, Abcam, 1:1000) antibodies were used as the primary antibodies. Anti-mouse IgG HRP-linked antibody (7076, Cell signaling Technology, 1:2000) and anti-rabbit IgG HRP-linked antibody (7074, Cell signaling Technology, 1:2000) were used as the secondary antibodies. A list of the antibodies used is also provided in Supplementary Data 2.

### ChIP experiments
After treatment with HU or CPT for 80 min, HeLa/Fucci2 cells were crosslinked with 1% formaldehyde. For ChIP experiments using anti-

DHX9, the chromatin fraction was isolated by subcellular fractionation from HeLa cells before crosslinking. After quenching with 0.125 M glycine, cells were lysed with SDS lysis buffer (1% SDS, 10 mM EDTA, 50 mM Tris−HCl pH 8.0) containing 1× complete protease inhibitor cocktail (Roche, Switzerland) and sonicated with a Bioruptor UCD-300 (Cosmobio, Japan). After centrifugation, the supernatants were diluted in a 9-fold volume of ChIP dilution buffer (1.1% Triton X-100, 167 mM NaCl, 0.11% DOC, 50 mM Tris−HCl pH 8.0) and precleared with Dynabeads Protein G (Thermo Fisher Scientific). Aliquots of the chromatin lysate were incubated overnight at 4 °C with antibodies or IgG. The immunocomplex was recovered by Dynabeads Protein G (Thermo Fisher Scientific), which were blocked with 0.5% bovine serum albumin (BSA) and 100 µg/ml salmon sperm DNA. The beads were then washed and reverse crosslinked. DNA was purified with phenol−chloroform extraction and subjected to RT-qPCR analysis. anti-E2F1 (ab179445, Abcam, UK, 2 µg), anti-E2F6 (ab53061, Abcam, 2 µg), anti-cMyc (5605, Cell Signaling Technology, MA, USA, 1:500), and Rabbit IgG (PM035, MBL, Japan, 2 µg) antibodies were used. Primer sets used are shown in Supplementary Data 2.

## Super-resolution single-molecule fluorescence microscopy (Stochastic Optical Reconstruction Microscopy, STORM)

For immunofluorescence combined with the smFISH assay, ViewRNA Cell Plus Assay Kit (Thermo Fisher Scientific) was used according to the manufacturer's instructions. Spliced and intron-retained transcripts were detected using a probe set targeting exon 2, detecting both spliced and intron-retained transcripts (ViewRNA Probe Set; Assay ID: VA1-11879, Thermo Fisher Scientific), and a probe set targeting both intron 1 and intron 2, which can detect only intron-retained transcripts (ViewRNA Probe Set; Assay ID: VF6-6000434, Thermo Fisher Scientific). Super-resolution images were acquired on a single-molecule fluorescence microscope (HM-1000, NanoresoTM, Sysmex, Japan). The HM-1000 is single-molecule localization microscope that utilizes photoswitchable fluorescent dyes[83] and overcomes the diffraction limit of conventional confocal microscopy, with a resolution of about 20 nm. Samples labeled with self-blinking fluorescent dyes were continuously excited with an exposure time of 30 ms, and 20,000 frames were imaged. Super-resolution microscopy analyzes were conducted in triplicate throughout the experiments. Four to 10 cells were examined in each experiment. The center coordinates of each fluorescence unit were extracted by image analysis and superimposed to construct a super-resolution image[84]. AlexaFluor488, AlexaFluor546, and AlexaFluor647 channels were acquired in super-resolution mode. The laser conditions for each wavelength were as follows: 488 nm, 33.2 mW; 561 nm, 18.2 mW; 637 nm, 38 mW. The DAPI channel was acquired in fluorescence mode using a 405 nm laser (1.1 mW). Multicolor fluorescent beads (Multiple Fluorophore Fluorescent Particles, 0.1–0.3 µm diameter, FP-0257-2, Spherotech, IL, USA) were used to correct for chromatic aberration and drift between channels. 100 µL of diluted bead solution (1 drop in 5 mL PBS) was added dropwise to the sample coverslip, and the mixture was allowed to stand at room temperature for 5 min in the dark. After washing twice with 1 mL of PBS, the samples were mounted using an imaging buffer (Sysmex), according to the manufacturer's instructions. Super-resolution image construction and processing was conducted using ImageJ (Fiji 2.9.0), using the function "Analyze Particles" in ImageJ defining the contours of a particle at each wavelength. The numbers of particles and particles that overlapped with other particles in different wavelengths were counted automatically.

Anti-phosho RPA32 (Ser33) (A300-246A-M, BETHYL, 1:200), anti-RNA Helicase A (DHX9) (ab26271, Abcam, 1:200), anti-PCNA (HPA030521, Sigma−Aldrich, MO, USA 1:100), anti-phospho-Histone H2A.X (Ser139), Alexa Fluor 488, (05-636-AF488, Merck Millipore, MA, USA, 1:100), anti-phospho-Histone H2A.X (Ser139), and Alexa Fluor 647 (05-636-AF647, Merck Millipore, 1:100) were used as the primary

antibodies. Anti-rabbit IgG secondary antibody, Alexa Fluor 647 (A21244, Thermo Fisher Scientific, 1:1200) was used as the secondary antibody. The antibodies used are listed in Supplementary Data 2. For labeling of newly synthesized DNA by EdU incorporation, cells were incubated with 100 µM EdU for 20 min and then processed using Click-iT™ Plus EdU Alexa Fluor™ 647 Flow Cytometry Assay Kits (Thermo Fisher Scientific), according to the manufacturer's instructions.

## RIP and CLIP assays

For RIP assay, RNA in HeLa-RPA70-EGFP, HeLa-RPA32-EGFP, U2OS-RPA70-EGFP, and U2OS-RPA32-EGFP cells were pulled down by anti-GFP antibody (M048-3, MBL, 2 µg) or mouse IgG (I5381, Sigma−Aldrich, 2 µg), and analyzed by RT-qPCR[85]. For the CLIP assay, RNA-protein complexes were UV crosslinked and immunoprecipitated[86]. According to the protocol, RNase I digestion after lysis was performed. Anti-RNA Helicase A (DHX9) antibody (ab26271, Abcam, 2 µg), anti-phosho RPA32 antibody (Ser33) (A300-246A-M, BETHYL, 4 µg), or rabbit IgG (PM035, MBL, 2 µg or 4 µg) were used. A list of antibodies and primer sets are shown in Supplementary Data 2.

## Biotin-RNA pull-down

Biotinylated RNA bound to streptavidin beads pulled down associated proteins[87]. Biotin-labeled RNAs were in vitro transcribed using AmpliScribe T7-Flash Biotin-RNA transcription Kits (Lucigen, WI, USA). Nuclear lysates were prepared with Magna Nuclear RIP™ (Native) Nuclear RNA-Binding Protein Immunoprecipitation Kits (Sigma−Aldrich). RNA-protein complexes were recovered using Dynabeads MyOne Streptavidin T1 (Thermo Fisher Scientific). Pull-down protein samples were run on SDS-PAGE gels, followed by detection with Silver Stain KANTOIII (KANTO CHEMICAL, Japan) or Western blotting. The band around 140 kDa detected by the silver staining was excised and subjected to peptide digestion. Mass spectrometry (LC−MS/MS) analyzes of the digested peptides utilized LTQ Orbitrap Velos pro mass spectrometry (Thermo Scientific). Protein with an approximate molecular mass of 140 kDa detected in sense TUG1 pull-down product but not in antisense TUG1 pull-down product was identified as TUG1-interacting protein.

## CARPID assay

CARPID BASU-dCasRx was a gift from Jian Yan & Liang Zhang (Addgene plasmid # 153209). pXR004: CasRx pre-gRNA cloning backbone was a gift from Patrick Hsu (Addgene plasmid # 109054). HEK293T cells transfected with CARPID BASU-dCasRx and either pXR004-NT or pXR004-TUG1 (Supplementary Data 2) were subjected to the CARPID assay[36]. Briefly, cells were treated with 200 µM biotin and incubated for 15 min, then washed three times with cold PBS and lysed with 1 ml lysis buffer (50 mM Tris−HCl pH 7.4, 150 mM NaCl, 0.5% Triton X-100, 1 mM EDTA supplemented with protease inhibitors). After rotating at 4 °C for 20 min, supernatants were collected by centrifugation and quantified for normalization. Biotinylated proteins were recovered with Dynabeads MyOne Streptavidin T1. Proteins were eluted from the beads into elution buffer (2x SDS Sample Buffer, 8% 2-Mercaptoethanol, 5 mM Biotin) by incubation for 10 min at 95 °C. Plasmids, sgRNAs are shown in Supplementary Data 2.

## R-loop detection by slot blot

Total nucleic acid was extracted from cell nuclei using NucleoSpin Tissue kits (MACHEREY-NAGEL, Germany). Purified DNA was treated with or without 1 U of RNase H (New England Biolabs) overnight at 37 °C, and applied to positively charged Nylon Membranes (Roche) assembled in the BioDot-SF microfiltration apparatus (Bio-Rad, CA, USA) with TBS buffer (10 mM Tris pH 7.5, 150 mM NaCl). The blotted DNA was crosslinked to the membrane using Stratalinker UV Cross-linker 2400 (Stratagene, CA, USA) at 120 mJ/cm². Anti-DNA-RNA

Hybrid antibody, clone S9.6 (MABE1095, Merck Millipore, 1:1000) (Supplementary Data 2) was used for detecting RNA/DNA hybrids. To normalize the amount of DNA, the same membrane was stained with 1x SYBR™ Gold Nucleic Acid Gel Stain (Thermo Fisher Scientific) in TBS-T (TBS buffer with 0.1% Tween 20). After sequential washes with TBS-T and TBS thrice each, fluorescent images were captured by FUSION Chemiluminescence Imaging System with Spectral Capsule 480 and F535 filter (VILBER, France).

### DNA fiber analysis

4 h after ASO transfection, cells were first incubated with 25 μM chlorodeoxyuridine (CldU) and then with 250 μM iododeoxyuridine (IdU) for 15 min each. Cells are resuspended in PBS, dropped onto glass slides, and then lysed with DNA fiber lysis buffer[88]. The glass slides are tilted to extend DNA and then fixed with Carnoy fluid (MeOH:AcOH, 3:1) for 3 min, 70% EtOH for 1 h, and MeOH for 3 min. The slides are treated with HCl to denature DNA and then neutralized with sodium tetraborate[88]. The slides were then treated with rat anti-BrdU antibody (ab6326-250, Abcam, 1:150) and mouse anti-BrdU antibody (347580, BD Biosciences, NJ, USA, 1:500), which reacted against CldU and IdU, respectively. Cy3-conjugated anti-rat IgG (712-165-153, Jackson ImmunoResearch, PA, USA, 1:400) and Alexa Fluor 488 anti-mouse IgG (A11029, Thermo Fisher Scientific, 1:100) were used as the secondary antibodies (Supplementary Data 2). Fiber spreads were prepared from $0.5 \times 10^6$ cells/ml. Images were captured with a fluorescent microscope (DMI6000B, Leica) with 63x objectives using LAS X 3.3 (Leica). Fiber lengths were measured from fibers without overlay using ImageJ (Fiji 2.9.0), and micrometer values were expressed in kilobases using the following conversion factor: 1 μm = 2.59 kb. A list of the drugs is provided in Supplementary Data 2.

### Flow cytometry (FCM)

Cells were washed with PBS and incubated for 15 min on ice in hybridization buffer (PBS containing 1.0% BSA and 0.25% Triton X-100). After centrifugation, cells were hybridized with an anti-γ-H2AX antibody (05-636-AF488, Merck Millipore, 1:100) for 1 h in the dark at room temperature (24–26 °C). Cells were then stained with FxCycle™ Violet Stain (Thermo Fisher Scientific) for 30 min before FCM. An apoptosis assay was conducted by using Annexin V-FITC Apoptosis Detection Kit (Nacalai Tesque). For analysis of EdU incorporation into newly synthesized DNA, cells were incubated with 10 μM EdU for 1 h and then processed using Click-iT™ Plus EdU Alexa Fluor™ 647 Flow Cytometry Assay Kit (Thermo Fisher Scientific), according to the manufacturer's instructions. The stained samples were analyzed using a Gallios flow cytometer (Beckman Colter, CA, USA) using Kaluza for Gallios 2.0 (Beckman Colter) (Supplementary Fig. 12). The data were analyzed using FlowJo software 10.6.1 (BD Biosciencies). The percentage of each cell-cycle population was analyzed by ModFit LT 5.0 (Verity Software House, ME, USA).

### Tethering of TUG1 on LacO-repeat locus using LacO/LacR and PP7/PCP system

The experiments were performed as shown in Supplementary Fig. 5a. Briefly, U2OS 2-6-3/TA were co-transfected with pDisplay-mCherry-LacR-PCP and either pcDNA3.4-TUG1-PP7, pcDNA3.4-Luciferase-PP7, or TUG1-deletion constructs (Δ1–4) (Supplementary Data 2). PP7 and PCP sequences were derived from Pcr4-12xMBS-PBS (Addgene plasmid # 52984) and ubc-nls-ha-MCP-VenusN-nls-ha-PCP-VenusC (Addgene plasmid # 52985), respectively (Supplementary Data 2). For tethering of DHX9 on the LacO-repeat locus (Supplementary Fig. 5d), pDisplay-mCherry-LacR-DHX9 (DHX9) or pDisplay-mCherry-LacR-DHX9 helicase-dead mutant (DHX9 mut), were transfected into U2OS 2-6-3/TA (Supplementary Data 2). Then the transfected U2OS 2-6-3/TA cells were cultured with or without 3 μg/ml of Dox for 24 h, followed by DRIP analysis. Cell images were captured with a

fluorescent microscope (DMI6000B, Leica) with 40x objectives using LAS X 3.3 (Leica).

### DRIP and library preparation for DRIP-seq

Genomic DNA containing RNA/DNA hybrids was extracted with Buffer-M (6 M guanidine thiocyanate, 10 mM Tris−HCl pH 6.5, 20 mM EDTA, 4% Triton X-100, 1% N-lauroylsarcosine, 1% DTT)[67]. Samples were sonicated to a peak fragment size of 250 bp, and treated with or without 1 U of RNase H overnight at 37 °C. After 20-fold dilution with DRIP buffer (50 mM Tris−HCl pH 8.0, 150 mM NaCl, 1 mM EDTA, 0.05% Triton X-100), samples were incubated with Anti-DNA-RNA Hybrid antibody, clone S9.6 (MABE1095, Merck Millipore, 1:100) for 12 h at 4 °C. Then Dynabeads Protein G beads were added for 2 h. Bound beads were washed three times in binding buffer and elution was performed in elution buffer (50 mM Tris pH 8.0, 10 mM EDTA, 0.5% SDS, 1.0 mg/ml Proteinase K) for 45 min with rotation at 55 °C. DNA was purified by phenol−chloroform extraction and ethanol precipitation. The products were analyzed by qPCR or subjected to DRIP-seq. For the latter, a paired-end library was generated using NEBNext Ultra DNA library prep kits (New England BioLabs) according to the manufacturer's instructions.

### DRIP-seq data processing

The sequencing reads were mapped to hg19 using STAR (version 2.5.3). Duplicated reads were then removed using MarkDuplicates.jar (Picard version 1.29), Peak-calling (MACS2, version 2.2.7.1) and IDR (irreproducible discovery rate, ide version 2.0.3) analyzes were performed according to the ENCODE guidelines[89,90]. Briefly, peak-calling was performed with a less stringent $p$-value threshold (1e-3), and peak consistency was evaluated based on signal values with a 1% threshold. The ratio between the number of peaks consistent between true replicates (Nt) and between pooled pseudoreplicates (Np) was calculated in all the combinations. When all the combinations among three replicates satisfy Np/Nt <2, this indicates reliable replicates. Peaks consistently satisfying the above criteria among replicates were used for downstream analyzes. Alterations in library size-normalized read count in peaks after ASO and/or CPT treatment were evaluated using DiffBind (version 3.0.15), and peak annotation was performed using homer (version 4.11.1). Metaplots were generated using the computeMatrix function in deepTools (version 3.5.1). The enrichment of repetitive elements was evaluated by intersecting peak regions with a UCSC RepeatMasker track using Bedtools (version 2.30.0). We also used γ-H2AX ChIP-seq datasets in HeLa cells in a previous study (GSE108172)[43].

### CARPID-based ChIP-qPCR

HEK293T cells transfected with CARPID BASU-dCasRx and either pXR004-NT or pXR004-TUG1 were incubated with 200 μM biotin for 15 min, then washed three times with cold PBS. Nuclei were collected by subcellular fractionation[82] and crosslinked with 1% formaldehyde. After quenching with 0.125 M glycine, cells were lysed with lysis buffer (50 mM Tris−HCl pH 7.4, 150 mM NaCl, 0.5% Triton X-100, 1 mM EDTA supplemented with protease inhibitors) and sonicated using Covaris S220 (Covaris Inc., MA, USA). After centrifugation, the supernatants were diluted 9-fold in lysis buffer. Biotinylated proteins and crosslinked DNA were recovered with Dynabeads MyOne Streptavidin T1. The beads were then washed and reverse crosslinked. DNA was purified by phenol−chloroform extraction and subjected to RT-qPCR analysis. Plasmids, sgRNAs, and primer sets used are shown in Supplementary Data 2.

### MSI analysis

HeLa/Fucci2 cells were transfected with either Ctrl ASO or TUG1#1 twice a week for two weeks. The MSI evaluation was performed by capillary electrophoresis, and MSI status is judged by visual

assessment of allele size change[91]. DNA was extracted using NucleoSpin Tissue kits and the microsatellite loci D2S123, D5S346 and D17S250 were amplified using Quick Taq HS DyeMix (TOYOBO). PCR products were analyzed on an ABI 310 Genetic Analyzer (Applied Biosystems, CA, USA) and the results were processed using GeneMapper v4.1 (Applied Biosystems) software. Primer sets are shown in Supplementary Data 2.

## Neutral comet assay
Neutral comet assay was performed following the manufacturer's instruction (Trevigen, MD, USA) with some modification. Cells were incubated with or without a caspase Inhibitor (20 µM Z-VAD-FMK, S7023, Selleck) for 1 h before the 24 h of ASO transfection. Cells were then embedded in Low-Melting Agarose, spread, and solidified over the Comet Slides on ice. The slides were immersed in Lysis Buffer for at least 1 h and then incubated in neutral electrophoresis buffer (0.1 M Tris–Ac pH 9.0, 0.3 M NaOAc·3$H_2$O). After electrophoresis for 1 h at 0.75 V/cm at 4 °C, samples were fixed in precipitation buffer (1 M $NH_4$Ac, 85% ethanol) and 70% ethanol. After staining nuclei by 1x SYBR™ Gold Nucleic Acid Gel Stain, images were randomly captured with a fluorescent microscope (DMI6000B, Leica) with 40× objectives. Images were analyzed by Comet Assay IV software (Perceptive Instruments, UK) using "Olivetail moment" as a parameter of the extent of DSB.

## Cell proliferation assay
A total of $2.5 \times 10^4$ cells per well were seeded in 6-well plates. After 24 h, cells were transfected with ASO. Cells were harvested by trypsinization at different time points, and the numbers of cells were determined by manual counting. For statistical analysis, global curve-fitting by nonlinear regression (Exponential Malthusian growth) was performed using GraphPad Prism 9.4.1 software. The best-fit $k$ value (the rate constant) was calculated for each data set (the growth of cells treated with Ctrl ASO, TUG1#1, and TUG1#2), carried out in triplicate. One-way ANOVA and Tukey's multiple comparison tests were used for the statistical analysis.

## Pyrosequencing analysis
500 ng of genomic DNA was bisulfite converted using an Epitect Plus bisulfite kit (Qiagen). *MGMT* promoter region was amplified by PCR and DNA methylation levels of 3 CpGs were measured by pyrosequencing (PyroMark Q24 system, Qiagen)[92]. Primer sequences used are shown in Supplementary Data 2.

## Xenograft mouse brain tumor model and treatment
Animal protocols were approved by the Animal Care and Use Committee of Nagoya University Graduate School of Medicine (approval number 20271). Mice were housed under standard, regulated conditions; 12/12 light/dark cycle, temperature at 21 °C ± 4 °C and humidity 40–70%. LN229 cells ($1 \times 10^5$ per mouse) were injected intracranially into 6-week-old female NOD/SCID mice (The Jackson Laboratory Japan). Branched PEGylated poly-(l-ornithine) (PEG-PLO) was used as the DDS of ASOs in vivo[24,48,93]. Two weeks after the injection, CTRL-DDS (1 mg/kg of CTRL ASO per day) or antiTUG1-DDS (1 mg/kg of TUG1#1 per day) were intravenously injected every 3 days for 30 days; TMZ (2.5 mg/kg per day) was intraperitoneally injected one day before CTRL-DDS or antiTUG1-DDS treatment. U251MG cells ($5 \times 10^4$ per mouse) were also injected into 6-week-old female NOD/SCID mice. One week after the injection, TMZ (2.0 mg/kg per day) was intraperitoneally injected, followed by two days of CTRL-DDS (1 mg/kg of CTRL ASO per day) or antiTUG1-DDS (1 mg/kg of TUG1#1 per day) treatment. The three day treatment schedule was repeated four times. ASOs used are shown in Supplementary Data 2. MRI images were evaluated to determine tumor mass area and volume using Horos 3.0 software (Horos Project, MD, USA). After treatment, brain tissue was harvested

and examined histologically. For this, brain tissues were fixed for 48 h in 10% neutral buffered formalin and embedded in paraffin, sections were made (5 µm-thick), stained with hematoxylin and eosin (HE). Samples were serially sectioned until the maximum tumor area was visible (44–370 sections).

## Abbreviations
A list of abbreviations is given in Supplementary Table 5.

## Reporting summary
Further information on research design is available in the Nature Portfolio Reporting Summary linked to this article.

## Data availability
The DRIP-seq data generated in this study have been deposited in the Gene Expression Omnibus (GEO) under accession code DRA013393. The microarray data and analyzed DRIP-seq data have been deposited in the Genomic Expression Archive (GEA) under accession codes E-GEAD-362 and E-GEAD-488, respectively. The human cancer data from the cancer genome atlas (TCGA) are derived from GEPIA (Gene Expression Profiling Interactive Analysis) (http://gepia.cancer-pku.cn). Source data are provided with this paper.

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

## Acknowledgements

This study was supported by the Japan Agency for Medical Research and Development (23ck0106816h0001, 22ama221204h0001, 21ck0106566h0001, Y.K.; 19cm0106108h0004, Y.K.; and 19cm0106202s0404, M.M.S.), and performed as research programs of the Grant-in-Aid for Scientific Research, the Japan Society for the Promotion of Science (20H03511, 17H03582, Y.K.; 20K07570, 17K07247, M.M.S.). We thank Shinichi Tatsumi and Susumu Sawaguchi (Sysmex) for super-resolution image acquisition, Takuya Abe and Kouji Hirota (Tokyo Metropolitan University) for help with the DNA fiber technique, and Reiko Nakagawa (Laboratory for Phyloinformatics at the RIKEN Center for Biosystems Dynamics Research (BDR)) for conducting mass spectrometry analysis. We also thank David L. Spector (Cold Spring Harbor Laboratory) for providing U2OS 2-6-3 cells and Minoru Takata (Kyoto University) valuable insights into the work. The authors wish to acknowledge the Division for Medical Research Engineering, Nagoya University Graduate School of Medicine, for technical support. Part of this study was conducted through the Joint Usage/Research Center Program of the Radiation Biology Center, Kyoto University.

## Author contributions

M.M.S., K.I., and Y.Ko designed and directed the project. M.M.S and K.I. performed most of the experiments and analyzed the data with assistance from K.S., Y.M, J.X, X.W., Y.Kit, A.M., Y.Kib, T.N., F.O., R.S., S.S., J.K., R.Y, K.M and K.K. K.O. and H.S. conducted DRIP-seq data analysis. M.M.S., K.I. and Y.Ko wrote the manuscript. All authors discussed the results and commented on the manuscript.

## Competing interests

The authors declare no competing interests.
