## [Peer Review File · Nature Communications]

TUG1-mediated R-loop resolution at microsatellite loci as a prerequisite for cancer cell proliferationREVIEWER COMMENTS

Reviewer #1 (Remarks to the Author):

In this manuscript, the authors investigated the long non-coding RNA, TUG1, and found that it resolves R-loops, and interacts with RPA and DHX9 helicase at certain loci with specific repeats, in cancer cells. Depletion of TUG1 results in replication stress (RS), accumulation of R-loop and DNA damage, and apoptosis. Mouse xenograft experiments showed that a combination therapy with anti-TUG1-1DDA and TMZ may be a promising therapy for glioblastoma. The findings are potentially interesting. The authors performed numerous experiments, and most were well-conducted. The paper is well written mostly. However, some critical experiments and explanations are missing, and interpretations are sometimes confusing. Several issues below should be addressed.

Major comments:

1. There is a difference of FISH data in the most part of this paper and their previous paper. Their single molecule FISH data in this paper gives an impression that most TUG1 is in the nucleus. However, a regular FISH in Figure 7G and data in their previous paper (Katsushima et al, Nature Commun, 2016, <https://www.nature.com/articles/ncomms13616>) show that substantial TUG1 molecules are in the cytoplasm. What makes the difference? The authors should clarify the cellular distribution of TUG1. They should make a direct comparison of the nuclear and cytoplasmic TUG1 in their subcellular fractionation, in Fig.1D. Also, they should be careful about interpretation of direct and indirect cellular effects under the TUG1 knockdown.
2. For Fig. 5, it would be nice if the authors can show TUG1 is bound the TUG1 KD sensitive site, by Biotin-RNA pull down-qPCR.
3. If TUG1 is overexpressed in the CPT-treated cells, and responsible for R-loop resolution, is the R-loop level reduced in the CPT-treated cells? If it is not the case, as the authors briefly describe in the discussion on page 17, line 1, or in another paper (Marinello et al, Nucleic Acids Res, doi:10.1093/nar/gkt778), the authors should explain in more detail. Also, the authors should show that R-loop is hyper-augmented in cells co-treated with TUG1-ASO and CPT, by slot blot. DRIP-SEQ data in Fig. 5 C-D are somewhat confusing in this aspect.
4. Overall, it may help to have a model-figure to show the up/down stream events: RS (by CPT/HU treatment), RS (by TUG1 depletion), ATR-CHK1, DNA damage, R-loop formation, DHX9/RPA binding, R-loop resolution, and apoptosis. It is confusing, because RS is caused by CPT/HU treatment (page 6, "...induced RS by treatment with hydroxyurea (HU) or camptothecin (CPT)", where TUG1 is overexpressed. In addition, RS is also caused by TUG1 depletion (Fig. 7A, Summary." Depletion of TUG1 lead to overabundant R-loops and enhanced RS.").
5. In Fig. 7H, do the three genes have the CA repeat? Are the repeats important in mouse as well?

Minor comments:

1. The summary starts with "Oncogene-induced DNA replication stress (RS)...". There is no experiment clearly and directly addressing the oncogene-induced DNA RS in this manuscript, therefore, it should be re-written.
2. On page 16, the second from the last line: "Supplementary Fig. S1J" is not presented in the current manuscript.
3. A paper from Dr. Rinn's research group was published in Nat Commun, describing that nuclear

localization of the TUG1 lnc RNA is driven by intron retention (Dumbovic et al, Nat Commun, 2021, <https://doi.org/10.1038/s41467-021-23221-w>). In the present paper, TUG1 cDNA with no intron was used to map the site of interactions with DHX9 and RPA32 in the nucleus. The authors should discuss about the possibility of the intronic sequence that is bound to TUG1.

4. It is reported that DHX9 is localized in the nucleolus functionally (Thacker et al, Sci Rep, 2020, <https://doi.org/10.1038/s41598-020-75160-z>, Leone et al, EMBO Rep, <https://www.ncbi.nlm.nih.gov/pmc/articles/PMC5494521/>) . Please comment on your TUG1-FISH and DHX9 IF data on this aspect. Did you see the nucleolar localization, in a population of cells?

Reviewer #2 (Remarks to the Author):

In the manuscript entitled „TUG1-mediated R-loop resolution at microsatellite loci as a prerequisite for cancer cell proliferation“, Suzuki and Kondo claim that upregulation of TUG1 resolved the problem of R-loop accumulation during DNA replication stress and led to increased tumor growth. In general, the individual data points should be displayed. There should be a quantification of WBs. The author should also highlight the biological replicate of experiments. The experimental models are inconsistent (for example, in Fig 2, U2OS was investigated, and in Fig. 3, HeLa was used). The author should explain why the various cell lines were used for different experiments. Too many abbreviations were used, and the author did not provide the full name of some of the abbreviations. The gating of cell cycle experiments should be presented in all figures. There are plenty of visualizations. However, from my point of view, some simple tables with useful statistics are more informative. The claim that targeting TUG1 coupled with DNA damaging agents provides a novel strategy for the treatment of cancers is not well experimentally supported, since only one cancer cell line was investigated in *in vivo* treatment. The author studies the function of lncRNA. Which transcriptional variant of TUG1 did they focus on? Especially in a publication from Dumbovic et al. shows that TUG1 has a bimodal distribution of fully spliced cytoplasmic and intron-retained nuclear transcripts.

1. Fig. 1: They only used mitotic shake-off to obtain synchronized cell cultures. They should also use retrospective synchronization achieved by flow sorting to validate their finding. Especially to better detect and characterize the cells in S-Phase, the author should use BrdU or Edu staining for accurately measuring DNA replication. The investigated FxCycle Violet staining is only for bivariate analysis of total DNA content.
2. Fig. 1A shows the fold change of lncRNA expression after two hours of treatment. How many DMSO samples were used as a control?
3. Fig. S1A: could the author explain what mVenus-positive means?
4. Fig. S1A: could the author provide the statistical analysis of FACS data?
5. Fig. 1D: are all these data statistics non-significant? The author could use the subcellular fraction isolation and rt-qPCR to validate smFISH results.
6. Fig. S1F: how many times was siRNA KD processed? Since there is only one western blot figure without statistics, it looks like only one time KD was processed.
7. Fig. 1G: since ATR inhibitor VE-821 is investigated, WB of ATR expression should be added.
8. Fig. S1H showed the comparison between tumor and normal. Could the author clarify what is expected under the definition of normal cells (normal brain? Blood cells?...)
9. Fig. 2B: how did the author analyze co-localization? By counting. How many times did the author process the experiment? It looks like the investigation was processed only once, and multiple nuclei were counted.
10. Fig. 2C and E, where is the corresponding quantification?
11. Fig. S2A, more information about plasmids and developing stable overexpression should be provided in the method section.
12. Fig 3B, please provide the quantification of replicas.
13. Fig. 3E: in the main text, Fig. 3F should be changed to Fig. 3E. for the conclusion, the author

needs to calculate the ratio since the input protein amount is inconsistent.

14. Fig. 4B: Why the DRB treatment was investigated? The author should explain the purpose of this experiment.

15. Fig. 4C: the experiment was repeated three times. Therefore, it should be only three data points in the figure. It is incorrect to show all 900 fibers and process the static analysis using 900 data points.

16. Fig. S4G: it should be a separate table demonstrating the percentage of different R-loop regions.

17. Fig. 6A and B: did the experiment process only one time? Please provide the static analysis.

18. Fig.6D: please provide the results of TUG1#2 as well.

19. Fig. 6E. which statistical analysis was used? It is significant only for 7 days.

20. Fig. 6G: increased H2AX percentage of cells doesn't directly indicate the increase of apoptosis.

21. Fig. 7: patients with GBM containing a methylated MGMT promoter benefited from TMZ treatment. What is the methylation status of MGMT in LN229?

Reviewer #3 (Remarks to the Author):

Suzuki and colleagues report that the taurine upregulated gene InRNA plays a role in the resolution of replication stress induced R-loops and propose that its downregulation might be employed as a potent therapeutic approach for treating cancer.

While a role for TUG1 in R-loop resolution would be novel, the case made here is not compelling. Nor was it proposed how TUG1 facilitates R-loop resolution other than proposed interaction with DHX9 that one assumes must be distinct from the mechanism through which DHX9 resolves other R-loops.

Specific criticisms:

1. Fig 1D plots the relative number of TUG1 something (molecules? Foci?) but it does not say relative to what (time 0?). The values for nuclear TUG1 in S/G2 cells treated with HU go up and down but no stats are provided. How do the numbers of TUG1 foci relate to the numbers of TUG1 molecules at sites of replication stress. Presumably the authors can measure RS by quantifying stalled DNA Polymerase or stalled RNA Polymerase?

2. Fig 1G shows a contribution of ATR pathway signalling to the RS induced increase in TUG1 expression. Why is ATR/pATR not included in the western blot? Similarly why were the other key DNA damage induced signalling pathways (involving ATM/ DNA-PK) not examined, especially as Fig 6 references Chk2 a downstream target of ATM?

3. A major criticism of this manuscript is the reliance on co-localization data. It is not clear what co-localization means here. Only few representative images are shown, often with foci false coloured with red and green. Colocalization should at least in some way be detected as a yellow colour resulting from the superimposed green and red foci. These are not that abundant with many green and red foci not colocalised and more often than not colocalization is only partial. More importantly, colocalization,, particularly when indicative of a physical interaction as suggested in the manuscript for TUG1 and DHX9, would result not only in a coincidence of the foci (i.e. in the same place) but also a co-distribution (i.e. the shape and density of the two signals would vary co-ordinately). This does not seem to be the case. How was colocalization measured, what were the criteria used in scoring (by eye or by computer analysis?) and how was it analysed? This is critical if these data are to establish a compelling case for proper colocalization. Numerical data is not presented for all the colocalization studies (e.g. Fig 2E).

4. The interaction between TUG1 and RPA is performed after overexpression of RPA subunits (Fig 2D). However the authors already made the point that their observations are cell cycle dependent and the over-expression of RPA will disrupt this. The experiment should rely on the endogenous properly regulated RPA proteins

5. In Fig 2E the authors use a catalytically dead form of RNaseH-GFP to bind DNA:RNA hybrid. This

tool is likely to detect hybrid generated as replication primers as well as R-loops. How much of the signal they measure is due to this? Does the signal go away if they knock down primase in the cells?

6. Depletion of TUG1 reduces the speed of replication elongation by a small but seemingly significant amount. What happens to replication fork progression if cells are induced for RS between the periods of labelling? i.e. in response to DNA damage treatment? The images in Suppl Fig 3C are not high quality as DNA fibres appear overlaid and difficult to measure. There seems to be an issue with combing.

7. Fig 4D suggests that the cell cycle is prolonged in cells depleted of TUG1 and that this is due to an extended S-phase. It would be preferable to see FACS plots of PI and BrdU incorporation as this gives a better measure of ongoing nucleotide incorporation and the completion of replication.

Notwithstanding the lower replication progression in cells lacking TUG1, do the authors know that replication delay is caused by slower replication fork progression or whether the replicative issues cause a proportion of cells to arrest/ die during S-phase? The latter would fit with the data reported later showing proliferation assays for normal v TUG1 depleted cells. This might also be supplemented by survival curves in response to the RS inducing agents used in the study.

8. The authors label their cell cycle plots as cells that are 2n and 4n, which indicates diploid and tetraploid. Do they mean 2c and 4c which is a measure of diploid DNA content (non-replicated v replicated genomes)

9. Fig 4G - the authors propose that 'data indicated that TUG1 resolves R-loops via DHX9, which is induced by transcription'. This is an overstatement as it is based solely on a correlation.

10. The role of DHX9 in R loop metabolism is currently unclear with competing assessments of its role in R-loop resolution (DOI: 10.1016/j.celrep.2018.04.025 and DOI: 10.1038/s41467-018-06677-1). In both papers however, knockdown of DHX9 resulted in a decrease in global R-loops. Therefore it cannot necessarily be inferred that an interaction with DHX9 might promote R-loop resolution. More analysis is required including genomic approaches with depleted DHX9 and or expression of DHX9 helicase defective mutant.

11. The DRIP-seq experiments seem to be the strongest data and speak directly to the different localization and quantity of R-loops

12. The authors report that loss of TUG1 induces DNA damage and apoptosis. They measure damage in the form of DNA breaks with the phosphorylated form of histone H2AX or directly as a tail moment. What proportion of these DSB are specific RS induced DNA breaks and not DSB caused by fragmentation during apoptosis?

13. The authors suggest that treatment of glioblastoma cells with TMZ and depletion of TUG1 has a synergistic effect. While there is an increased effect this does not at all look like synergy, perhaps additive though.

Overall there are a number of issues to be addressed before the current manuscript gets close to establishing its hypothesis. At the moment the manuscript is structured as if the authors are trying to show a role for TUG1 in R-loop resolution, rather than critically evaluating whether or not it does.

We thank all three reviewers for their constructive comments. In order to address these important comments, we have performed many more experiments and found that the new data consistently support the original findings.

Reviewer #1:

In this manuscript, the authors investigated the long non-coding RNA, TUG1, and found that it resolves R-loops, and interacts with RPA and DHX9 helicase at certain loci with specific repeats, in cancer cells. Depletion of TUG1 results in replication stress (RS), accumulation of R-loop and DNA damage, and apoptosis. Mouse xenograft experiments showed that a combination therapy with anti-TUG1-1DDA and TMZ may be a promising therapy for glioblastoma. The findings are potentially interesting. The authors performed numerous experiments, and most were well-conducted. The paper is well written mostly. However, some critical experiments and explanations are missing, and interpretations are sometimes confusing. Several issues below should be addressed.

Thank you for the constructive comments. In order to address these important comments, we have performed additional experiments.

Major comments:

1. There is a difference of FISH data in the most part of this paper and their previous paper. Their single molecule FISH data in this paper gives an impression that most TUG1 is in the nucleus. However, a regular FISH in Figure 7G and data in their previous paper (Katsushima et al, Nature Commun, 2016, <https://www.nature.com/articles/ncomms13616>) show that substantial TUG1 molecules are in the cytoplasm. What makes the difference? The authors should clarify the cellular distribution of TUG1. They should make a direct comparison of the nuclear and cytoplasmic TUG1 in their subcellular fractionation, in Fig.1D. Also, they should be careful about interpretation of direct and indirect cellular effects under the TUG1 knockdown.

TUG1 is generally located both in the nucleus and cytoplasm, as was reported in previous studies including ours (Cabili et al., 2015 *Genome Biol.* DOI: 10.1186/s13059-015-0586-4, Katsushima et al., 2016, *Nature Commun*). We found that newly transcribed TUG1 appeared particularly in the nucleus two hours after either CPT or HU treatment, followed by increased TUG1 molecules in the cytoplasm at four hours (Fig. 1C, 1D). The single molecule FISH data in Fig. 1C are representative images showing nuclear localization of TUG1 two hours after CPT or HU treatments (Fig. 1C, 1D). As suggested, we further performed a direct comparison of the nuclear and cytoplasmic TUG1 in subcellular fractions using qRT-PCR (Supplementary Fig. S1D).

As for direct and indirect cellular effects, we studied cells after four hours of TUG1 knockdown (direct effect) by Slot blot (Fig. 4B), DNA fiber assay (Fig. 4C), and DRIP-seq (Fig. 5). These experiments clearly showed that TUG1 KD induced R-loop accumulation and slowed fork progression (RS) within this time frame. Subsequently, after 24 hours of TUG1 KD, we observed both direct and indirect effects as a consequence of accumulated unresolved R-loops and RS, such as an increase of γ -H2AX signals (Fig. 6A, 6B), DNA fragmentation (Fig. 6C), suppressed cell growth (Fig. 6D), and apoptosis (Fig. 6E). We have now explained that a short period of ASO treatment is useful to understand the direct effects of TUG1 KD in the Results section.

2. For Fig. 5, it would be nice if the authors can show TUG1 is bound the TUG1 KD sensitive site, by Biotin-RNA pull down-qPCR.

Thank you for this helpful suggestion. In order to show TUG1 binding to the TUG1 KD-sensitive site, we conducted CARPID-based ChIP-qPCR, because this assay represents the detection of a physiological interaction of TUG1 with the target sites. This assay showed that TUG1 and proteins in its proximity were significantly enriched at the TUG1-sensitive regions (BCL2 and C22orf34) (Supplementary Fig. S6J). We have added these data to Supplementary Fig. S6J, Results, and the Materials and Methods sections.

3. If TUG1 is overexpressed in the CPT-treated cells, and responsible for R-loop resolution, is the R-loop level reduced in the CPT-treated cells? If it is not the case, as the authors briefly describe in the discussion on page 17, line 1, or in another paper (Marinello et al, Nucleic Acids Res, doi:10.1093/nar/gkt778), the authors should explain in more detail. Also, the authors should show that R-loop is hyper-augmented in cells co-treated with TUG1-ASO and CPT, by slot blot. DRIP-SEQ data in Fig. 5 C-D are somewhat confusing in this aspect.

We thank the reviewer for this important comment. In our DRIP-seq data, we recognized that the level of R-loop induced by co-treatment with TUG1-ASO and CPT is lower than with CPT treatment alone. This is probably due to a combination of TUG1 KD and 2 hours of CPT treatment decreasing cellular metabolism including transcription and DNA replication, resulting in suppression of overall R-loop formation in a certain population of cells. Marinello et al. demonstrated that CPT induces a transient increase of R-loops at the divergent active promoters. Because microsatellite tandem repeats are abundant in human promoters (Sawaya et al., PLoS One 2013, DOI: 10.1371/journal.pone.0054710), TUG1 may function to resolve R-loops formed at the promoter regions with microsatellite repeats. We have added sentences to this effect to the Results.

4. Overall, it may help to have a model-figure to show the up/down stream events: RS (by CPT/HU treatment), RS (by TUG1 depletion), ATR-CBK1, DNA damage, R-loop formation, DHX9/RPA binding, R-loop resolution, and apoptosis. It is confusing, because RS is caused by CPT/HU treatment (page 6, "...induced RS by treatment with hydroxyurea (HU) or camptothecin (CPT)", where TUG1 is overexpressed. In addition, RS is also caused by TUG1 depletion (Fig. 7A, Summary. "Depletion of TUG1 lead to overabundant R-loops and enhanced RS. ").

We are sorry for any confusion in our manuscript. As suggested, we have now added a model figure as Supplementary Fig. S11.

5. In Fig. 7H, do the three genes have the CA repeat? Are the repeats important in mouse as well?

BCL2, C22orf34, and PRS6KA2 genes that harbor CA repeats in the intronic regions are localized in TUG1-sensitive loci by the DRIP-seq analysis. In Fig. 7H, we assessed the abundance of R-loop in these gene regions in human glioma cells, LN229, taken from the xenograft mouse model. The function of TUG1 on R-loop resolution at the CA repeats in mouse cells is not clear because we unfortunately haven't studied this thus far. In the revised version, we have added sentences to this effect in the figure legend of Fig. 7H.

Minor comments:

1. The Summary starts with "Oncogene-induced DNA replication stress (RS)···". There is no experiment clearly and directly addressing the oncogene-induced DNA RS in this manuscript, therefore, it should be rewritten.

We corrected the first sentence accordingly in the Summary.

2. On page 16, the second from the last line: "Supplementary Fig. S1J" is not presented in the current manuscript.

We sincerely apologize for this and have corrected it.

3. A paper from Dr. Rinn's research group was published in Nat Commun, describing that nuclear localization of the TUG1 lncRNA is driven by intron retention (Dumbovic *et al.*, Nat Commun, 2021, <https://doi.org/10.1038/s41467-021-23221-w>). In the present paper, TUG1 cDNA with no intron was used to map the site of interactions with DHX9 and RPA32

in the nucleus. The authors should discuss about the possibility of the intronic sequence that is bound to TUG1.

Thank you for this comment. This point was also raised by reviewer #2. Dumbovic *et al.* showed that both fully spliced and intron-retained transcripts are present in the nucleus, while only spliced transcripts exist in the cytoplasm of HeLa cells. In the revised manuscript, we examined spliced and intron-retained transcripts by smFISH using a probe set targeting exon 2, which detects both spliced and intron-retained transcripts (ViewRNA Probe Set; Assay ID: VA1-11879, ThermoFisher), and a probe set targeting both intron 1 and intron 2, which can detect only intron-retained transcripts (ViewRNA Probe Set; Assay ID: VF6-6000434, ThermoFisher). After CPT treatment, R-loops (RNH1^{D210N}-GFP) colocalized with both fully spliced and intron-retained transcripts. These data suggest that exon regions are sufficient for the resolution of R-loops. However, further work is needed to define the functional difference between fully spliced and intron-retained transcripts. We have added these data and sentences to Supplementary Fig. S2D, Material and Methods, Results, and Discussion.

4. It is reported that DHX9 is localized in the nucleolus functionally (Thacker *et al*, *Sci Rep*, 2020, <https://doi.org/10.1038/s41598-020-75160-z>, Leone *et al*, *EMBO Rep*, <https://www.ncbi.nlm.nih.gov/pmc/articles/PMC5494521/>). Please comment on your TUG1-FISH and DHX9 IF data on this aspect. Did you see the nucleolar localization, in a population of cells?

We observed the localization of neither TUG1 nor DHX9 to the nucleolus in the S phase in our current experimental setting, in which we used two antibodies against DHX9 (ab26271, Abcam, and SC-137232, Santa Cruz) (Figure 1 in this point-by-point response). Unfortunately, the work of Thacker *et al.* cannot be exactly reproduced because the antibody which they used in their study is no longer available (SC-66997, Santa Cruz). It might be possible that DHX9 can be recruited to the nucleolus in certain cells under certain conditions.

Figure 1. Representative image of a HeLa cell co-stained with DAPI (blue), nucleolin (green), and DHX9 (red). DHX9 localizes to the nucleoplasm. Antibodies used were rabbit polyclonal anti-nucleolin (A300-711A, BETHYL) and mouse monoclonal anti-NDH II (DHX9) (sc-137232, Santa Cruz). Scale bar, 10 μ m.

Reviewer #2

In the manuscript entitled „TUG1-mediated R-loop resolution at microsatellite loci as a prerequisite for cancer cell proliferation”, Suzuki and Kondo claim that upregulation of TUG1 resolved the problem of R-lop accumulation during DNA replication stress and led to increased tumor growth. In general, the individual data points should be displayed. There should be a quantification of WBs. The author should also highlight the biological replicate of experiments. The experimental models are inconsistent (for example, in Fig 2, U2OS was investigated, and in Fig. 3, HeLa was used). The author should explain why the various cell lines were used for different experiments. Too many abbreviations were used, and the author did not provide the full name of some of the abbreviations. The gating of cell cycle experiments should be presented in all figures. There are plenty of visualizations. However, from my point of view, some simple tables with useful statistics are more informative. The claim that targeting TUG1 coupled with DNA damaging agents provides a novel strategy for the treatment of cancers is not well experimentally supported, since only one cancer cell line was investigated in in vivo treatment. The author studies the function of lncRNA. Which transcriptional variant of TUG1 did they focus on? Especially in a publication from Dumbovic et al. shows that TUG1 has a bimodal distribution of fully spliced cytoplasmic and intron-retained nuclear transcripts.

O-1. *In general, the individual data points should be displayed.*

Thank you for this comment. In the revised version, we have displayed data points on bar graphs.

O-2. *There should be a quantification of WBs.*

Thank you for this comment. In the revised version, we quantified the band densities of the Western blots and performed statistical analyses.

O-3. *The author should also highlight the biological replicate of experiments.*

We apologize for this omission. We have now added information on the biological replicates of experiments in the Figure legends of the revised manuscript.

O-4. *The experimental models are inconsistent (for example, in Fig 2, U2OS was investigated,*

and in Fig. 3, HeLa was used). The author should explain why the various cell lines were used for different experiments.

We agree with this comment. To consistently perform experiments using HeLa cells, we have added data from newly established HeLa-RPA32-GFP and HeLa-RPA70-GFP cell lines for the RIP experiments in Fig. 2D. Now HeLa cells were used throughout the *in vitro* study. We also used U2OS cells in addition to HeLa cells particularly in DNA damage-related experiments because U2OS has been well characterized and widely used in the analysis of DNA repair due to its intact DNA damage response. Finally, we strove to apply our new findings more to the clinical context, particularly for glioma, a representative orphan disease and refractory cancer. For this, glioma cell lines LN229, T98G and U251MG, were investigated to assess the effectiveness of targeting TUG1 as a model for the treatment of glioma, in addition to some of the key experiments related to TUG1 function *in vitro*. We have added these data and sentences to Fig. 2D, the Materials and Methods, and the Results.

O-5. Too many abbreviations were used, and the author did not provide the full name of some of the abbreviations.

We apologize for this inconvenience. We rechecked the text and made sure that abbreviations are fully spelled out when they first appear in the text. In addition, a list of abbreviations was also added as Supplemental Table S8.

O-6. The gating of cell cycle experiments should be presented in all figures.

We agree with this comment. In the revised version, cell cycle gating by ModFit LT 5.0 software is presented.

O-7. There are plenty of visualizations. However, from my point of view, some simple tables with useful statistics are more informative.

Thank you for this comment. According to the suggestion, in addition to the super-resolution microscopy visualizations, we added a summary of the results to Supplementary Table S2.

*O-8. The claim that targeting TUG1 coupled with DNA damaging agents provides a novel strategy for the treatment of cancers is not well experimentally supported, since only one cancer cell line was investigated in *in vivo* treatment.*

We agree with this comment. We additionally conducted therapy with antiTUG1-DDS and TMZ in the U251MG glioma xenograft mouse model. This combination therapy most effectively suppressed U251MG tumor growth, as was also seen in LN229 glioma cells. We added these data and sentences to Supplementary Fig. S10 G-I, the Materials and Methods, and the Results sections.

O-9. The author studies the function of lncRNA. Which transcriptional variant of TUG1 did they focus on? Especially in a publication from Dumbovic et al. shows that TUG1 has a bimodal distribution of fully spliced cytoplasmic and intron-retained nuclear transcripts.

Thank you for this comment. This point was also raised by reviewer #1. Dumbovic *et al.* showed that both fully spliced and intron-retained transcripts are present in the nucleus, while only spliced transcripts exist in the cytoplasm of HeLa cells. In the revised manuscript, we examined spliced and intron-retained transcripts by smFISH using a probe set targeting exon 2, which detects both spliced and intron-retained transcripts (ViewRNA Probe Set; Assay ID: VA1-11879, ThermoFisher), and a probe set targeting both intron 1 and intron 2, which can detect only intron-retained transcripts (ViewRNA Probe Set; Assay ID: VF6-6000434, ThermoFisher). After CPT treatment, R-loops (RNH1^{D210N}-GFP) colocalized with both fully spliced and intron-retained transcripts. These data suggest that exon regions are sufficient for the resolution of R-loops. However, further work is needed to define the functional difference between fully spliced and intron-retained transcripts. We have added these data and sentences to Supplementary Fig. S2D, Material and Methods, Results, and Discussion.

1. Fig. 1: They only used mitotic shake-off to obtain synchronized cell cultures. They should also use retrospective synchronization achieved by flow sorting to validate their fining. Especially to better detect and characterize the cells in S-Phase, the author should use BrdU or Edu staining for accurately measuring DNA replication. The investigated FxCycle Violet staining is only for bivariate analysis of total DNA content.

Thank you for this important comment. As pointed out, we synchronized the HeLa/Fucci2 cells by mitotic shake off and used the cells for analysis after 13 hours. In order to better detect and characterize the cells in S-Phase, we used EdU staining for accurately measuring DNA replication along with FxCycle Violet-based flow cytometric analysis (Supplementary Fig. S1A). We validated the finding that after 13 hours of mitotic shake off, EdU-positive cells

(i.e., in S phase) were significantly enriched by this method (>90%). We have added these data and sentences to Supplementary Fig. S1A, Materials and Methods, and the Results.

2. Fig. 1A shows the fold change of lncRNA expression after two hours of treatment. How many DMSO samples were used as a control?

We are sorry for this inadequate explanation. Cells were treated with either DMSO, HU, or CPT and examined in duplicate. We have corrected these sentences in Materials and Methods and Supplementary Table S1.

3. Fig. S1A: could the author explain what mVenus-positive means?

HeLa/Fucci2 cells utilize the fluorescent ubiquitination-based cell cycle indicator (Fucci) system, which visualizes cell cycle progression in live cells; G1 cells are mCherry-positive (shown in red), and S/G2 cells are mVenus-positive (shown in green) (Sakaue-Sawano *et al.*, 2011, *BNC Cell Biol*, DOI: 10.1186/1471-2121-12-2). In the original manuscript, mVenus-positive cells were assessed to show enrichment of S-phase cells 13 hours after mitotic shake off. However, mVenus-positive cells contain both S and G2 cells. In the revised manuscript, we detected cells with DNA replication by EdU incorporation experiments in order to precisely label S-phase cells according to the reviewer's suggestion (please see comments #1 and #4). We added sentences to Materials and Methods, Results, and figure legend of Supplementary Fig. S1A.

4. Fig. S1A: could the author provide the statistical analysis of FACS data?

This comment is related to your comment #1. In the revised manuscript, we showed that after 13 hours of mitotic shake off, a majority of cells (>90%) was EdU-positive (S phase). We have added the statistical analysis to Supplemental Fig. S1A and explanatory sentences to Materials and Methods.

5. Fig. 1D: are all these data statistics non-significant? The author could use the subcellular fraction isolation and rt-qPCR to validate smFISH results.

Thank you for this suggestion. We have statistically analyzed the data and added the results to Fig. 1D. We have also validated the smFISH results with RT-qPCR after isolating subcellular fractions in Supplementary Fig. S1D.

6. Fig. S1F: how many times was siRNA KD processed? Since there is only one western blot figure without statistics, it looks like only one time KD was processed.

We apologize for this insufficient presentation. We have performed this series of experiments in triplicate. We have now added quantifications of the gel images with statistical analysis to Supplementary Fig. S1G (originally Fig. S1F) in the revised version.

7. Fig. 1G: since ATR inhibitor VE-821 is investigated, WB of ATR expression should be added.

We agree with this comment. The Western blotting of p-ATR and ATR has been added to Fig. 1G.

8. Fig. S1H showed the comparison between tumor and normal. Could the author clarify what is expected under the definition of normal cells (normal brain? Blood cells?...)

We apologize for the inadequate information. The seven tumors and corresponding normal tissues in TCGA and the GTEx dataset are as follows: DLBC: Lymphoid Neoplasm Diffuse Large B-cell Lymphoma and Blood; GBM: Glioblastoma multiforme and Brain; PAAD: Pancreatic adenocarcinoma and Pancreas; THYM: Thymoma and Blood; CHOL: Cholangiocarcinoma and Bile ducts; LGG: Brain Lower Grade Glioma and Brain; LAML: Acute Myeloid Leukemia and Bone Marrow. We have added this information to the figure legend of Supplementary Fig. S1I (originally Fig. S1H) in the revised version.

9. Fig. 2B: how did the author analyze co-localization? By counting. How many times did the author process the experiment? It looks like the investigation was processed only once, and multiple nuclei were counted.

This is an important point and was also raised by reviewer #3. We have analyzed super-resolution images using ImageJ software. This software contains a function "Analyze Particles" that extracts the contours of the particle at each wavelength. Then we automatically counted the number of particles and particles that overlapped with other particles in different colors. We have added this information to the Materials and Methods. We have conducted the super-resolution microscopy analysis in triplicate. Four to 10 cells were examined in individual experiments. Fig. 2B, 2C, and 2E show data from a representative single

experiment. The independent experimental data are summarized in Supplementary Table S2. We added information on the biological replicates and the number of examined cells to the figure legends.

10. Fig. 2C and E, where is the corresponding quantification?

We have added the quantitative analysis of the co-localization to Fig. 2C and 2E.

11. Fig. S2A, more information about plasmids and developing stable overexpression should be provided in the method section.

We have added this information to the Materials and Methods.

12. Fig 3B, please provide the quantification of replicas.

We have added quantification of the Western blotting signals to Fig. 3B.

13. Fig. 3E: in the main text, Fig. 3F should be changed to Fig. 3E. for the conclusion, the author needs to calculate the ratio since the input protein amount is inconsistent.

We sincerely apologize for this typo. We also quantitatively analyzed the Western blots throughout the manuscript including Fig. 3E.

14. Fig. 4B: Why the DRB treatment was investigated? The author should explain the purpose of this experiment.

Thank you for this comment. DRB is a CDK9 inhibitor that rapidly arrests transcription initiation. It has been reported that DRB treatment strongly reduces R-loops (Sanz *et al.*, Mol Cell, 2016, DOI: 10.1016/j.molcel.2016.05.032). In our study, the DRB treatment is a control experiment showing that the signal detected by the S9.6 antibody represents R-loops formed by transcription. We have added a sentence to the Results section accordingly.

15. Fig. 4C: the experiment was repeated three times. Therefore, it should be only three data points in the figure. It is incorrect to show all 900 fibers and process the static analysis using 900 data points.

Thank you for this comment. According to the suggestion, we now show the data from a single representative experiment from four cell lines in Fig. 4C. In addition, each triplicate from the DNA fiber assay is summarized in Supplementary Table S4. The statistical analysis is added for the independent experiments.

16. Fig. S4G: it should be a separate table demonstrating the percentage of different R-loop regions.

Thank you for this comment. We have added the data to Supplementary Table S5 in addition to Supplementary Fig. S6F (originally Fig. S4G).

17. Fig. 6A and B: did the experiment process only one time? Please provide the static analysis.

We quantified band densities of the Western blot gel images, which were obtained from three independent experiments (i.e., in triplicate), and are shown as a bar graph with statistics (Fig. 6A). We also performed flow cytometry in triplicate and statistically analyzed the data (Fig. 6B).

18. Fig. 6D: please provide the results of TUG1#2 as well.

We have added neutral comet assay data using TUG1#2 ASO to Supplementary Fig. S8D.

19. Fig. 6E. which statistical analysis was used? It is significant only for 7 days.

In the original manuscript, we conducted statistical analyses only on day seven using the Student's t-test. In the revised version, we performed statistical analyses on all the proliferation curves. We performed global curve-fitting by nonlinear regression analysis (Exponential Malthusian growth) using GraphPad Prism 9.4.1 software. The best-fit k (the rate constant) value was calculated for each data set (the growth of cells treated with Ctrl ASO, TUG1#1, and TUG1#2), carried out in triplicate. One-way ANOVA and Tukey's multiple comparison tests were newly used for statistical analysis. We have added a sentence to the Materials and Methods and new statistics to Fig. 6D (Fig. 6E in the original version).

20. Fig. 6G: increased H2AX percentage of cells doesn't directly indicate the increase of apoptosis.

We agree with this comment. Accordingly, we have added the data showing that 48 hours of TUG1 depletion did not induce apoptosis in TIG3 to Fig. 6F.

21. Fig. 7: patients with GBM containing a methylated MGMT promoter benefited from TMZ treatment. What is the methylation status of MGMT in LN229?

Thank you for this important comment. DNA methylation levels in the MGMT promoter in LN229 cells are very high (82.7% by pyrosequencing analysis). Consistent with this, LN229 is highly sensitive to TMZ treatment (less than 30 μ M) (Figure 2 in this point-by-point response). We examined another glioblastoma cell line, U251MG, which shows a lower level of DNA methylation in the MGMT promoter region (23.2%) and is less sensitive to TMZ treatment (around 100 μ M) (Figure 2 in this point-by-point response). In the revised manuscript, we investigated the effects of TUG-ASO with/without TMZ treatment in both LN229 and U251MG cells *in vivo*. As we show in Fig. 7F and Supplementary Fig. S10 G-I, this combination therapy significantly decreased the tumor size compared to TMZ alone in mouse xenografts of both LN229 and U251 cell lines. We have added the data and a sentence to Supplementary Fig. S10F, Materials and Methods, and Results sections.

Figure 2. TMZ sensitivity assay for glioblastoma cell lines used in the mouse xenograft experiments. Relative growth rates for LN229 (left) and U251MG (right) treated with TMZ. Cells were seeded at 1×10^3 cells/well into 96-well plates and treated with different concentrations of TMZ for 7 days. Cell viability was assessed using Cell Count Reagent SF (Nacalai Tesque, Japan).

Reviewer #3

*Suzuki and colleagues report that the taurine upregulated gene *lnRNA* plays a role in the resolution of replication stress induced R-loops and propose that its downregulation might be employed as a potent therapeutic approach for treating cancer.*

While a role for TUG1 in R-loop resolution would be novel, the case made here is not compelling. Nor was it proposed how TUG1 facilitates R-loop resolution other than proposed interaction with DHX9 that one assumes must be distinct from the mechanism through which DHX9 resolves other R-loops.

Thank you for these important comments. To the extent possible, these have been addressed by the addition of new data.

Specific criticisms:

1. Fig 1D plots the relative number of TUG1 something (molecules? Foci?) but it does not say relative to what (time 0?). The values for nuclear TUG1 in S/G2 cells treated with HU go up and down but no stats are provided. How do the numbers of TUG1 foci relate to the numbers of TUG1 molecules at sites of replication stress. Presumably the authors can measure RS by quantifying stalled DNA Polymerase or stalled RNA Polymerase?

The results of single molecule FISH in Fig. 1D showed a value relative to the median number of TUG1 molecules in the S/G2 nucleus at 0 h. We apologize for this and have now corrected the figure legends and added the significance assessments.

We thank the reviewer for this insightful question about TUG1 foci and RS sites. After CPT treatment, around half of TUG1 foci colocalized with R-loops (RNH1^{D210N}-GFP). Proliferating cell nuclear antigen (PCNA), which is a co-factor of DNA polymerase-delta (Bravo *et al.*, Nature, 1987, DOI: 10.1038/326515a0) showed striking colocalization with about half of the TUG1/RNH1^{D210N}-GFP foci (Supplementary Fig. S2B) (Please also see comment #5). We also labeled newly synthesized DNA with EdU to visualize the locations where replication occurs. TUG1 foci also colocalize with EdU signals (Supplementary Fig. S2C) (Please also see comment #5). In view of the fact that TUG1 foci colocalize with pRPA32 (Fig. 2A), these data indicate that TUG1 is located at the RS site, which consists of a stalled fork and an R-loop. We have added the new data and a sentence to Supplementary Fig. S2C, Fig. S2D, Materials and Methods, Results, and Discussion sections.

*2. Fig 1G shows a contribution of ATR pathway signalling to the RS induced increase in TUG1 expression. Why is ATR/pATR not included in the western blot? Similarly why were the other key DNA damage induced signalling pathways (involving ATM/ DNA-PK) not examined, especially as Fig 6 references *Chk2* a downstream target of ATM?*

We agree with this comment. The Western blotting of p-ATR/ATR and p-ATM/ATM has been added to Fig. 1G. and Supplementary Fig. S8C, respectively.

3. A major criticism of this manuscript is the reliance on co-localization data. It is not clear what co-localization means here. Only few representative images are shown, often with foci false coloured with red and green. Co-localization should at least in some way be detected as a yellow colour resulting from the superimposed green and red foci. These are not that abundant with many green and red foci not colocalised and more often than not co-localization is only partial. More importantly, co-localization,, particularly when indicative of a physical interaction as suggested in the manuscript for TUG1 and DHX9, would result not only in a coincidence of the foci (i.e. in the same place) but also a co-distribution (i.e. the shape and density of the two signals would vary co-ordinately). This does not seem to be the case. How was co-localization measured, what were the criteria used in scoring (by eye or by computer analysis?) and how was it analysed? This is critical if these data are to establish a compelling case for proper co-localization. Numerical data is not presented for all the co-localization studies (e.g. Fig 2E).

Thank you for this criticism. This important point was also raised by reviewer #2. STORM microscopy has a high resolution of less than 40 nm, based on detecting single photoswitchable fluorophores in a labeled sample in time and precisely localizing them at much higher optical resolution than can be attained by confocal imaging systems (more than 200 nm). We employed super-resolution microscopy to investigate molecular interactions between TUG1 and proteins. Because the position and distribution of the secondary fluorescent-labeling molecules is different in RNA FISH and immunostaining, RNA FISH signals and protein IF signals do not have the same shape even if the primary molecules are bound. Please see other examples of a super-resolution microscopy image of lncRNA NEAT1 and NONO (Figure 3 in this point-by-point response). NEAT1 and NONO are well known to directly bind (Yamazaki *et al.*, 2018, Mol Cell, DOI: 10.1016/j.molcel.2018.05.019).

Based on the aforementioned idea, we have analyzed super-resolution images using ImageJ software. This software contains a function "Analyze Particles" that extracts the contours of the particle at each wavelength. Then the number of particles and particles that overlapped with other particles in different colors are automatically counted. We have added this information to the Materials and Methods. Quantification of the co-localization analysis in Fig. 2C and 2E has also been added. Finally, in addition to the super-resolution microscopy visualizations, we added a summary of the results to Supplementary Table S2.

Figure 3. Super-resolution image of NEAT1 (red) and NONO (green). Super-resolution image of NEAT1 (probe set; ViewRNA Probe Set; Assay ID: VA1-12621, ThermoFisher) and NONO (Rabbit polyclonal anti-NONO, A300-582A, BETHYL) in a HeLa cell. Inset panels are magnified regions of the main panel. Scale bar, 5 μ m.

4. The interaction between *TUG1* and RPA is performed after overexpression of RPA subunits (Fig 2D). However the authors already made the point that their observations are cell cycle dependent and the over-expression of RPA will disrupt this. The experiment should rely on the endogenous properly regulated RPA proteins.

Thank you for this comment. According to this suggestion, we have tested the four antibodies against RPA70 and four against RPA32 listed below, but none of these antibodies worked in the RIP assay. However, although our RIP experiments used overexpressed RPA subunits, we detected a consistent interaction between *TUG1* and endogenous RPA32 by the alternative CARPID technology *in vivo* (Supplementary Fig. S3D). We also validated the interaction between endogenous pRPA32 and *TUG1* by CLIP assay (Fig. 3D). Therefore we believe that *TUG1* does indeed interact with RPA.

Antibody name	Company	Catalog number
Anti-Replication Protein A (RPA70-9)	Calbiochem	NA13
RPA 70 kDa subunit (B-6)	Santa Cruz	sc-28304
RPA70 antibody	Novus	NB100-2204
Anti-RPA70 antibody [EPR3472]	abcam	ab79398
Anti-Replication Protein A (RPA34-19)	Calbiochem	NA18
Anti-RPA32/RPA2 antibody [EPR2877Y]	abcam	ab76420
RPA2 Rabbit Monoclonal Antibody (JE45-59)	ThermoFisher	MA5-34843
RPA32/RPA2 (E8X5P) XP® Rabbit mAb	CST	35869

5. In Fig 2E the authors use a catalytically dead form of RNaseH-GFP to bind DNA:RNA hybrid. This tool is likely to detect hybrid generated as replication primers as well as R-loops. How much of the signal they measure is due to this? Does the signal go away if they knock down primase in the cells?

Thank you for this comment. According to the suggestion, we knocked down the primases PRIM1 and PRIM2, together with RNH1^{D210N}-GFP expression. However, unfortunately, these treatments proved lethal to the cells, despite testing several different conditions. In response to comment #1, we showed that TUG1/RNH1^{D210N}-GFP foci colocalize with PCNA and EdU, indicating that TUG1 is located at stalled forks (Supplementary Fig. S2B, S2C). Consistent with this, RNH1^{D210N}-GFP spots were clearly increased by CPT treatment, which induces R-loops (Fig. 2E). As the reviewer commented, RNaseH is involved in removal of RNA primers from Okazaki fragments during DNA replication. However, the length of RNA primers is generally short, about 10-20 nt. Therefore, the signal intensity of RNaseH-GFP at the RNA primers might be weaker relative to that at R-loops, the length of which is generally more than a few kb.

6. Depletion of TUG1 reduces the speed of replication elongation by a small but seemingly significant amount. What happens to replication fork progression if cells are induced for RS between the periods of labelling? i.e. in response to DNA damage treatment? The images in Suppl Fig 3C are not high quality as DNA fibres appear overlaid and difficult to measure. There seems to be an issue with combing.

According to this comment, we strove to improve the combing process and count only those fibers that did not overlay (Fig. 4C and Supplementary Fig. S4C). Measurements with new imaging validated the original findings. We added a sentence to this effect to the Materials and Methods. Treatment with the DNA damaging agent CPT causes a significant slowdown of replication fork progression (Seiler *et al.*, 2007, Mol Cell Biol, DOI: 10.1128/MCB.02278-06, Vujanovic *et al.*, 2017 Mol Cell, DOI: 10.1016/j.molcel.2017.08.010). Likewise, TUG1 depletion induces DNA damage caused by accumulated R-loops, reducing the speed of replication elongation (please also see the following comment #7). In the revised version, we now explain the cause of the slowdown of replication fork progression by TUG1 depletion in the Results section.

7. Fig 4D suggests that the cell cycle is prolonged in cells depleted of TUG1 and that this is due to an extended S-phase. It would be preferable to see FACS plots of PI and BrdU incorporation as this gives a better measure of ongoing nucleotide incorporation and the

completion of replication. Notwithstanding the lower replication progression in cells lacking TUG1, do the authors know that replication delay is caused by slower replication fork progression or whether the replicative issues cause a proportion of cells to arrest/ die during S-phase? The latter would fit with the data reported later showing proliferation assays for normal v TUG1 depleted cells. This might also be supplemented by survival curves in response to the RS inducing agents used in the study.

Thank you for this comment. Four hours after TUG1 knock down (ASO transfection), we had shown R-loop accumulation (Slot blot in Fig. 4B and DRIP-seq in Fig. 5) and slowed fork progression (DNA fiber assay in Fig. 4C) in the original manuscript. According to the above suggestion, we further performed FCM analysis of DNA content and EdU incorporation after four hours of TUG1 KD (Supplementary Fig. S4D-G). The FCM analysis showed slight but significant reduction of EdU incorporation at this time (Supplementary Fig. S4D, Fig. S4E). Consequently, the cells showed an increase of γ -H2AX signals from 6 hours after TUG1 KD (Fig. 6A), DNA fragmentation at 24 h (Fig. 6C), and apoptosis at 48 h (Fig. 6E). Consistent with this, after 24 hours of TUG1 KD, EdU incorporation was dramatically decreased (Supplementary Fig. S4F, Fig. S4G). These data indicate that replication delay is mainly caused by slowed replication fork progression in the early response to TUG1 depletion (i.e. after four hours), but at a later time point, in addition to the replication delay, a proportion of cells undergo arrest and death during S phase, caused by DNA damage, that contributes to suppressing cell growth (Fig. 6D).

8. The authors label their cell cycle plots as cells that are 2n and 4n, which indicates diploid and tetraploid. Do they mean 2c and 4c which is a measure of diploid DNA content (non-replicated v replicated genomes)

Thank you for this helpful comment. We have corrected the labels in Fig. 4D, 6B, 6F, Supplementary Fig. S1A, S4D, S4H, S8B accordingly.

9. Fig 4G - the authors propose that 'data indicated that TUG1 resolves R-loops via DHX9, which is induced by transcription'. This is an overstatement as it is based solely on a correlation.

Thank you for this comment. We carefully revised this sentence in the Results.

10. The role of DHX9 in R loop metabolism is currently unclear with competing assessments of its role in R-loop resolution (DOI: 10.1016/j.celrep.2018.04.025 and DOI: 10.1038/s41467-018-06677-1). In both papers however, knockdown of DHX9 resulted in a

decrease in global R-loops. Therefore it cannot necessarily be inferred that an interaction with DHX9 might promote R-loop resolution. More analysis is required including genomic approaches with depleted DHX9 and or expression of DHX9 helicase defective mutant.

Thank you for this important comment. As pointed out, DHX9 has been reported to help R-loop formation when there is a lack of splicing factors (Chakraborty *et al.*, 2018). Cristini *et al.* demonstrated that siDHX9 locally increased R-loops at transcribed gene regions; however, it decreased the total amount of R-loops in the cells. These findings indicate that the role of DHX9 in R-loop metabolism is context-dependent, as the reviewer suggested. We first investigated the activity of DHX9 on R-loop resolution using the LacO-LacR system. When mCherry-LacR-DHX9 was tethered to the LacO locus, the accumulation of R-loops was significantly reduced (Supplementary Fig. S5D-F). In contrast, tethering the catalytically dead mutant (D511A, E512A) for DHX9-helicase activity (Lee *et al.*, 2014, J Biol Chem, DOI: 10.1074/jbc.M114.568535) did not suppress R-loop formation (Supplementary Fig. S5F). Our original findings showed that tethering TUG1 to the LacO locus significantly reduced R-loop accumulation (Fig. 4F), while tethering the deletion mutant of TUG1-PP7, which DHX9 cannot bind, did not suppress R-loop formation (Fig. 4G). Furthermore, depletion of DHX9 increased the amount of R-loops in this experimental setting (Fig. 4H). Consistent with this, we found that depletion of DHX9 increased R-loops at the TUG1-sensitive CA repeat regions BCL2 and C22orf34 (Supplementary Fig. S5F). These data strongly suggest that DHX9 resolves R-loops in an RNA helicase activity-dependent manner at certain loci. Indeed, the biochemical activity of DHX9 in R-loop resolution has also been reported by Dutta *et al.* (Dutta *et al.*, Methods in Molecular Biology, 2022, DOI: 10.1007/978-1-0716-2477-7_20). Taken together, these data indicate that DHX9 interacts with TUG1 and resolves R-loops via its catalytic activity at certain loci. As the reviewer suggested, R-loops at different loci seem to be resolved by different mechanisms. We have added these data and sentences to Supplementary Fig. S5D-F, Fig. S6F, Results and Materials and Methods sections.

11. The DRIP-seq experiments seem to be the strongest data and speak directly to the different localization and quantity of R-loops

Thank you for this encouraging comment. Our DRIP-seq data indicated that TUG1 specifically functions to resolve R-loops in CA repeat regions. In the revised version, we further report that DHX9 KD increased R-loop accumulation in TUG1-sensitive loci (Supplementary Fig. S6I). We propose that the TUG1-RPA-DHX9 interaction is a novel lncRNA-mediated mechanism for regulating R-loops at certain loci. We have added sentences to that effect to the Discussion.

12. The authors report that loss of TUG1 induces DNA damage and apoptosis. They measure damage in the form of DNA breaks with the phosphorylated form of histone H2AX or directly as a tail moment. What proportion of these DSB are specific RS induced DNA breaks and not DSB caused by fragmentation during apoptosis?

This is an important point. In the original manuscript, we carefully excluded cells at a late apoptosis stage from the measurements in the neutral comet assay by ignoring small comet heads and enlarged tails. In the revised version, we also depleted TUG1 with/without inhibition of apoptosis by the caspase inhibitor, Z-VAD-FMK, for 24 hours, and then performed neutral comet assays. No significant difference in the tail moment between cells treated with or without Z-VAD-FMK was observed (Supplementary Fig. S8E). Therefore, we conclude that the DSBs we detected were mostly induced by RS, and not by apoptosis. We have added these data and a sentence to Supplementary Fig. S8E, Materials and Methods, and Results sections.

13. The authors suggest that treatment of glioblastoma cells with TMZ and depletion of TUG1 has a synergistic effect. While there is an increased effect this does not at all look like synergy, perhaps additive though.

Thank you for this comment. In order to precisely assess synergistic effects, we further calculated the drug combination index (CI) for TUG1 ASO and TMZ treatment of LN229 and U251 glioma cell lines. The CI indicated a synergistic effect between TUG1 ASO and TMZ treatments. We have now added these data to Supplementary Fig. S9B-D, Materials and Methods, and Results sections.

Overall there are a number of issues to be addressed before the current manuscript gets close to establishing its hypothesis. At the moment the manuscript is structured as if the authors are trying to show a role for TUG1 in R-loop resolution, rather than critically evaluating whether or not it does.

Thank you for the constructive comments. We now believe that our new data convincingly support the original findings.

REVIEWER COMMENTS

Reviewer #1 (Remarks to the Author):

The authors satisfactory addressed all of the points I raised.

There is only one subtle point left, which is that they missed including the label "Supplementary Figure 11" in a list on the cover page of their supplementary materials file.

Basically, I suggest that the revised version is eligible for acceptance.

Reviewer #2 (Remarks to the Author):

The reversed paper thoroughly answered all my questions. Significantly, the author strongly updated the quality of all figures with suitable statistical analyses and clear explanations. The reversed version of the manuscript presents a high-quality science that meets Nature communication's quality.

Reviewer #3 (Remarks to the Author):

The authors have responded to specific points raised. My responses are below.

1. Fig 1D plots the relative number of TUG1 something (molecules? Foci?) but it does not say relative to what (time 0?). The values for nuclear TUG1 in S/G2 cells treated with HU go up and down but no stats are provided. How do the numbers of TUG1 foci relate to the numbers of TUG1 molecules at sites of replication stress. Presumably the authors can measure RS by quantifying stalled DNA Polymerase or stalled RNA Polymerase?

The results of single molecule FISH in Fig. 1D showed a value relative to the median number of TUG1 molecules in the S/G2 nucleus at 0 h. We apologize for this and have now corrected the figure legends and added the significance assessments.

We thank the reviewer for this insightful question about TUG1 foci and RS sites. After CPT treatment, around half of TUG1 foci colocalized with R-loops (RNH1D210N-GFP). Proliferating cell nuclear antigen (PCNA), which is a co-factor of DNA polymerase-delta (Bravo et al., Nature, 1987, DOI: 10.1038/326515a0) showed striking colocalization with about half of the TUG1/RNH1D210N -GFP foci (Supplementary Fig. S2B) (Please also see comment #5). We also labeled newly synthesized DNA with EdU to visualize the locations where replication occurs. TUG1 foci also colocalize with EdU signals (Supplementary Fig. S2C) (Please also see comment #5). In view of the fact that TUG1 foci colocalize with pRPA32 (Fig. 2A), these data indicate that TUG1 is located at the RS site, which consists of a stalled fork and an Rloop. We have added the new data and a sentence to Supplementary Fig. S2C, Fig. S2D, Materials and Methods, Results, and Discussion sections.

I am still not sure what the Y-axis is reporting. Relative number of TUG1 =15 means what. OK it is relative to the median but is this 15 fold, or 15 molecules or what?

It is still not clear how replication stress and particularly sites of replication stress is being defined. It has been shown that RNA-DNA hybrid per se does not stall replication so the colocalization of RNH1D210N, TUG1 and PCNA is not necessarily a site of replication stress. And as mentioned, the specificity of RNH1D210N as a measure of hybrid is not clear. It is not generally accepted that the most accurate way of detecting R-loops is through various ChIP methods with several different RNase controls and fluorescence based imaging methods should be backed up with other methods accordingly.

3. A major criticism of this manuscript is the reliance on co-localization data. It is not clear what co-localization means here. Only few representative images are shown, often with foci false coloured with red and green. Co-localization should at least in some way be detected as a yellow colour resulting from the superimposed green and red foci. These are not that abundant with many green and red foci not colocalised and more often than not colocalization is only partial. More importantly, co-localization,, particularly when indicative of a physical interaction as suggested in the manuscript for TUG1 and DHX9, would result not only in a coincidence of the foci (i.e. in the same place) but also a co-distribution (i.e. the shape and density of the two signals would vary co-ordinately). This does not seem to be the case. How was co-localization measured, what were the criteria used in scoring (by eye or by computer analysis?) and how was it analysed? This is critical if these data are to establish a compelling case for proper co-localization. Numerical data is not presented for all the colocalization studies (e.g. Fig 2E).

Thank you for this criticism. This important point was also raised by reviewer #2. STORM microscopy has a high resolution of less than 40 nm, based on detecting single photoswitchable fluorophores in a labeled sample in time and precisely localizing them at much higher optical resolution than can be attained by confocal imaging systems (more than 200 nm). We employed super-resolution microscopy to investigate molecular interactions between TUG1 and proteins. Because the position and distribution of the secondary fluorescent-labeling molecules is different in RNA FISH and immunostaining, RNA FISH signals and protein IF signals do not have the same shape even if the primary molecules are bound. Please see other examples of a super-resolution microscopy image of lncRNA NEAT1 and NONO (Figure 3 in this point-by-point response). NEAT1 and NONO are well known to directly bind (Yamazaki et al., 2018, Mol Cell, DOI: 10.1016/j.molcel.2018.05.019). Based on the aforementioned idea, we have analyzed super-resolution images using ImageJ software. This software contains a function "Analyze Particles" that extracts the contours of the particle at each wavelength. Then the number of particles and particles that overlapped with other particles in different colors are automatically counted. We have added this information to the Materials and Methods. Quantification of the co-localization analysis in Fig. 2C and 2E has also been added. Finally, in addition to the super-resolution microscopy visualizations, we added a summary of the results to Supplementary Table S2.

Obviously, a certain amount of image analysis needs to be taken on trust. However, the amount of co-localization in the graphs does not seem to align with visual inspection of the images. For example in Fig 2 none of the images look like there is 60% colocalization of pRPA with Tug1.

4. The interaction between TUG1 and RPA is performed after overexpression of RPA subunits (Fig 2D). However the authors already made the point that their observations are cell cycle dependent and the over-expression of RPA will disrupt this. The experiment should rely on the endogenous properly regulated RPA proteins.

Thank you for this comment. According to this suggestion, we have tested the four antibodies against RPA70 and four against RPA32 listed below, but none of these antibodies worked in the RIP assay. However, although our RIP experiments used overexpressed RPA subunits, we detected a consistent interaction between TUG1 and endogenous RPA32 by the alternative CARPID technology in vivo (Supplementary Fig. S3D). We also validated the interaction between endogenous pRPA32 and TUG1 by CLIP assay (Fig. 3D). Therefore we believe that TUG1 does indeed interact with RPA.

That is a pity. CARPID is a proximity labelling system and less convincing. Why would no RPA32 antibody work with RIP. RPA is not a particularly limiting protein.

5. In Fig 2E the authors use a catalytically dead form of RNaseH-GFP to bind DNA:RNA hybrid. This tool is likely to detect hybrid generated as replication primers as well as R-loops. How much of the signal they measure is due to this? Does the signal go away if they knock down primase in the cells?

Thank you for this comment. According to the suggestion, we knocked down the primases PRIM1 and PRIM2, together with RNH1D210N-GFP expression. However, unfortunately, these treatments proved lethal to the cells, despite testing several different conditions. In response to comment #1, we showed that TUG1/RNH1D210N -GFP foci colocalize with PCNA and EdU, indicating that TUG1 is located at stalled forks (Supplementary Fig. S2B, S2C). Consistent with this, RNH1D210N -GFP spots were clearly increased by CPT treatment, which induces R-loops (Fig. 2E). As the reviewer commented, RNaseH is involved in removal of RNA primers from Okazaki fragments during DNA replication. However, the length of RNA primers is generally short, about 10-20 nt. Therefore, the signal intensity of RNaseH-GFP at the RNA primers might be weaker relative to that at R-loops, the length of which is generally more than a few kb.

I am still not convinced of RNH1D210N -GFP as a tool, used in isolation of other robust mechanisms. I have seen knockdown of primases with RNH1D210N-GFP expression reported at a meeting where quite a lot of GFP signal was lost. I'm not sure if that study was published yet.

6. Depletion of TUG1 reduces the speed of replication elongation by a small but seemingly significant amount. What happens to replication fork progression if cells are induced for RS between the periods of labelling? i.e. in response to DNA damage treatment? The images in Suppl Fig 3C are not high quality as DNA fibres appear overlaid and difficult to measure. There seems to be an issue with combing.

According to this comment, we strove to improve the combing process and count only those fibers that did not overlay (Fig. 4C and Supplementary Fig. S4C). Measurements with new imaging validated the original findings. We added a sentence to this effect to the Materials and Methods. Treatment with the DNA damaging agent CPT causes a significant slowdown of replication fork progression (Seiler et al., 2007, Mol Cell Biol, DOI: 10.1128/MCB.02278-06, Vujanovic et al., 2017 Mol Cell, DOI: 10.1016/j.molcel.2017.08.010). Likewise, TUG1 depletion induces DNA damage caused by accumulated R-loops, reducing the speed of replication elongation (please also see the following comment #7). In the revised version, we now explain the cause of the slowdown of replication fork progression by TUG1 depletion in the Results section.

It is not clear for the fibre assay why the authors have used both CldU and IdU. The whole point of using two nucleotides is to incorporate a treatment between the two to see the effect of a treatment on fork speed etc. so that altered speed will be given by ratio of green over red. i.e. It is an internal control.

General issue

The authors wish to propose that TUG1 mediates R-loop resolution via DHX9. I am not wholly convinced by the data for the reasons previously outlined, and above. However, that aside, the authors do not present a model for how this occurs. Moreover, they rely on the interpretation that DHX9 unwinds R-loops as proposed by Cristini et al. However, most models for RDHX9 mediated unwinding of R-loops in the literature (over many years) are based on a 3'-5' unwinding activity whereas DHX9 is a 5'-3' helicase that requires a ssRNA tail for loading. ssDNA is not a good substrate for its activity. In other words DHX9 cannot act as an R-loop resolving protein as proposed and no other model has been put forward for how this might occur. Biochemically DHX9 was shown not to unwind canonical R-loops (i.e. with a 5' ssRNA tail). This conundrum needs to be resolved if the authors are to establish that TUG1 mediates R-loop resolution via DHX9.

REVIEWER COMMENTS

Reviewer #3 (Remarks to the Author):

The authors have provided some new data and several arguments in defence of their submission.

1. they have clarified what they are measuring and how they represent their data in Fig 1D. I am still not convinced by RNH1D210N-GFP as a tool, even though it has been used widely. S90.6 antibody was also widely accepted for use in immunofluorescence and has some utility but is now known to be much more reliable in robust ChIP experiments.

Thank you for the comment. We believe that the RNH1D210N-GFP signals that we observed specifically represent R-loops. However, we will consider performing ChIP with S9.6 antibody (DRIP) in future analysis.

2. The authors have provided an explanation for their co-localization data and new images. It is still not completely clear what co-localization here means. It seems that this method measures molecules in the close vicinity of each other but I don't think it is necessarily an indicator of direct (physical?) interaction.

We agree with your comments. Therefore, we further conducted other biochemical experiments to show direct interactions.

3. The inability to detect RPA in RIP is a pity.

We agree with your comment.

4. 4. See above (1) on RNH1D210N-GFP. It is a shame the Prim knockdown work is not published as it might have helped clear this issue up.

In our experimental setting, RNH1D210N-GFP did not seem to visualize RNA/DNA hybrids formed by RNA primers. Further studies will clarify whether RNH1 recognizes RNA primer regions, although it is out of the scope of our current study.

5. Fine

6. I don't think the authors provide any evidence for their mechanism for DHX9 function. The biochemistry of DHX9 polarity, substrate specificity etc suggests previous mechanisms proposed including that by Cristini cannot be correct. Moreover, Cristini presented data that despite their hypothesis that DHX9 resolves R-loops that knockdown of DHX9 reduced the global level of R-loops which would be more in line with data by Chakraborty et al Nat Comms 2018. It seems that a consensus mechanism for DHX9 is hard to establish. Unwinding of backtracked polymerase is unlikely as it would require dissociation of RNA Pol II. The authors have added a section to be more circumspect which is wise.

Thank you for this comment. We agree with the reviewer that a consensus mechanism for DHX9 is hard to establish at this stage.